# Transformer Fusion with Optimal Transport

**Moritz Imfeld**[*]**, Jacopo Graldi**[*]**, Marco Giordano**[*]**,**
**Thomas Hofmann, Sotiris Anagnostidis, Sidak Pal Singh**
ETH Zurich, Switzerland
{moimfeld, graldij, mgiordano}@ethz.ch

## Abstract

Fusion is a technique for merging multiple independently-trained neural networks in order to combine their capabilities. Past attempts have been restricted to the case of fully-connected, convolutional, and residual networks. This paper presents a systematic approach for fusing two or more transformer-based networks exploiting Optimal Transport to (soft-)align the various architectural components. We flesh out an abstraction for layer alignment, that can generalize to arbitrary architectures – in principle – and we apply this to the key ingredients of Transformers such as multi-head self-attention, layer-normalization, and residual connections, and we discuss how to handle them via various ablation studies. Furthermore, our method allows the fusion of models of different sizes (*heterogeneous fusion*), providing a new and efficient way to compress Transformers. The proposed approach is evaluated on both image classification tasks via Vision Transformer and natural language modeling tasks using BERT. Our approach consistently outperforms vanilla fusion, and, after a surprisingly short finetuning, also outperforms the individual converged parent models. In our analysis, we uncover intriguing insights about the significant role of soft alignment in the case of Transformers. Our results showcase the potential of fusing multiple Transformers, thus compounding their expertise, in the budding paradigm of model fusion and recombination. Code is available at https://github.com/graldij/transformer-fusion.

## 1 Introduction

Transformers, as introduced by Vaswani et al. (2017), have profoundly impacted machine learning, establishing a prevailing neural network architecture across various domains. Transformers consistently excel in different fields, including natural language processing (Lin et al., 2022), time series forecasting (Wen et al., 2022), and computer vision (Dosovitskiy et al., 2020). Their success can be attributed to their scaling properties (Kaplan et al., 2020) and efficient utilization of contemporary hardware architectures designed for extensive parallel computing. The unification of a single architecture across tasks facilitates immediate, far-reaching applicability of any analysis that handles general properties of the Transformer architecture.

As large Transformer foundation models (Bommasani et al., 2021) continue to grow in size and complexity, the challenges associated with training, i.e., exponential increase in parameters and compute for a fixed incremental improvement in performance (Hoffmann et al., 2022; Zhai et al., 2022; Bachmann et al., 2023), become increasingly more perilous. Consequently, achieving state-of-the-art results is often confined to researchers with access to ample GPU resources. To address these issues and strive for more efficient and sustainable performance improvements, we embark on the following more compelling and alternative inquiry:

*Can we combine the capabilities of pre-trained Transformer models?*

Merging multiple Transformer models into a single entity while preserving their unique capabilities can yield several advantages; (a) *Enhanced performance* by harnessing the collective capabilities of individual models. (b) *Reduced inference complexity*, as querying a single model replaces the need to query $n$ models in an ensemble, reducing computational (FLOPs) and storage requirements by

---

[*]These authors contributed equally to this work

a factor of $n$. (c) *The necessity to train from scratch can be readily eliminated*, leveraging existing public models, already available, and numerous in quantity [1].

A straightforward way of fusing, i.e., merging, models of the same architecture, is to average their weight matrices one-to-one, referred to as 'Vanilla Fusion' (VF). However, this method overlooks potential misalignments between the parameter matrices, arising due to neurons at the same positions, in different models, encoding different information (Godfrey et al., 2022). Instead, we propose to use Optimal Transport fusion (OTFusion) (Singh & Jaggi, 2020), which at its core, aligns the weight or parameter matrices before fusing them.

Thus, by virtue of such an alignment, OTFusion ensures that the fused model effectively integrates the knowledge and capabilities of the individual models to be merged, rather than simply averaging the weight matrices without guaranteeing meaningful information preservation. Additionally, OTFusion accommodates the fusion of models with different widths, and in turn, different sizes, which is fundamentally not possible with VF. This is a crucial feature, as such heterogeneous models are available in plenty, to better unleash the potential of existing pre-trained models. Consequently, OTFusion has been shown to be an effective method for fusing fully connected (Singh & Jaggi, 2020), convolutional (Nguyen et al., 2021) and recurrent neural networks (Akash et al., 2022) on a variety of tasks, heavily outperforming VF.

Yet, despite its wide adoption (Nguyen et al., 2021; Liu et al., 2022; Ainsworth et al., 2022), the layerwise procedure proposed by OTFusion does not fit well with contemporary architectural design, that comprises of constant residual streams, normalization layers, and attention operations. It is not equipped in any way to align and fuse models with complex information streams and to fuse transformer-specific components. Hence, the primary aim of our work is to develop techniques that help bridge these gaps and successfully generalize fusion to Transformer-based architectures.

**Our contributions are:** (a) We analyze each of the idiosyncratic architectural components in Transformers in thorough detail, with an ultimate aim to best fuse them across different models. Throughout our discussion, we exposit our approach based on the perspective of *flow of the transportation maps*[2], that makes for intuitive visualizations and interpretation. (b) We uncover that, surprisingly, OTFusion based on a *hard-alignment underperforms* in this context, contrary to the case of fully-connected or convolutional architectures; and that, *soft-alignment plays a key role* in successful one-shot fusion. (c) We showcase the efficacy of our approach by extensive experimentation involving the fusion and finetuning of Vision Transformers (ViTs) across multiple datasets, including CIFAR10, CIFAR100, TINY IMAGENET and IMAGENET-1K, as well as BERT (Devlin et al., 2018) models for natural language tasks. We *consistently outperform* the original *converged* models across tasks and datasets, by about $\sim$ *1.0%, while significantly reducing computational and storage costs by a factor of* $n$.

Overall, our research marks an important stride in advancing model fusion techniques, that help deliver enhanced performance and efficiency for modern Transformer based architectures.

## 2 RELATED WORK

**Model combination and ensembling.** The combination of multiple models has been a timeless idea in machine learning, from classical works on bagging and boosting (Breiman, 1996) to more contemporary approaches (Mienye & Sun, 2022; Garipov et al., 2018; Jolicoeur-Martineau et al., 2023). The key idea behind these works is to boost model performance, by capitalizing on the unique strengths of each model while mitigating their individual limitations. Or, more technically, one can think of model combination as a way of reducing the variance of the predictors (Geman et al., 1992). However, the main limitation is that such methods require the execution of each (parent) model for the final prediction, with a cost that scales linearly with the number of models.

**Model Fusion.** Model fusion (Singh & Jaggi, 2020; Wang et al., 2020; Wortsman et al., 2022; Matena & Raffel, 2022; Ainsworth et al., 2022; Nguyen et al., 2023) has emerged as a particularly notable direction in recent years, gaining significant traction in the machine-learning community. This line of work focuses on building better model combination approaches that account for the

---

[1] On huggingface there are more than 339,000 models available as of the 22nd of September 2023.

[2] This should be reminiscent of the flow of tensors in the computation graph of neural networks, and thus allows one to see a general strategy that can be potentially be adapted for any architecture type.

network structure and its inherent symmetries. We elaborate on some of these works, which are more relevant to the focus of our paper, below.

Singh & Jaggi (2020) propose a novel approach based on the OT theory exploiting the Wasserstein distance, where the neuron association allows fusing pre-existing models with the same depth in a *one-shot* fashion, thus without requiring retraining. OTFusion outperforms VF and was successfully used for model compression and fusion of CNNs, residual networks (ResNets), and multilayer perceptrons (MLPs). Since its publication, OTFusion has been extended in various ways. Nguyen et al. (2021) address the same-depth requirement of OTFusion. Liu et al. (2022) generalized the work as a graph-matching task, and taking into account the second-order similarity of model weights instead of linear alignment. Recent efforts on the topic have shown theoretical insights on fusion, extensions of previous algorithms to new network topologies, in particular, Akash et al. (2022) adapted OTFusion for recurrent networks, such as RNNs and LSTMs. Further, Stoica et al. (2023) propose an algorithm, for convolutional and residual architectures, that aims at finding redundant features within the same model and across the different models to be fused, so as to keep only meaningful and unique features in the fused model.

However, the fully layerwise interpretation of OTFusion (Singh & Jaggi, 2020) is currently only applicable to simple architectures such as MLPs, CNNs, and instances of ResNet. It is not equipped in any way to align and fuse models with complex information streams and to fuse transformer-specific components such as multi-head attention layers, layer-normalization, embeddings, or the sequential nature of the data.

**Fusion with a focus on Transformers.** Wortsman et al. (2022), in their approach of 'model soups', consider fusing transformer models that have a common backbone network that is pre-trained on the same dataset, but that are fine-tuned, say, with different hyperparameters. Owing to this, the models remain sufficiently close in the parameter space, which precludes the need to align them, and lets them employ just vanilla fusion (one-to-one averaging of the parameters) while still obtaining a gain in performance. Therefore, despite apparent practical gains, the 'model soup' approach is actually a poor representative of the complexity and intricacies of the general model fusion problem.

Arguably, the more empowering capability is to *fuse transformer networks that are potentially much more distant in their parameter spaces* and are diverse in nature. For instance, this arises when the networks have different initializations, or see examples in different batch orderings, or when they have different sizes, and more. This specific problem is tackled in this work, which is, to the best of our knowledge, *the first aiming at fusing transformer architectures by aligning their weights*.

**The conjecture of Linear Mode Connectivity (LMC) modulo permutations.** Given the recent interest around this conjecture posed in Entezari et al. (2021) and its wider demonstrations (Ainsworth et al., 2022), we would like to make a few clarifications: **(a)** The LMC barrier approaches zero only at **very high widths**, even for non-transformer architectures, see for instance Figure 4 of Ainsworth et al. (2022), and *importantly, not for any arbitrary width.* Thus, for typically sized residual or convolutional neural networks, the LMC barrier in loss is not zero at all, and the corresponding barrier when measured in accuracy is even more palpable. **(b)** Transformers possess **a more non-convex landscape**, as shown by Park & Kim (2022) in a comparison of vision transformers with residual networks, which consequently brings about higher LMC barriers. This can also be seen due to the fact that transformers contain components which further proliferate the number of symmetries, such as within- and across-head permutations as well as the translation invariance of softmax, — all of which serve to interfere the linear interpolation of parameters. Thus, the barriers in (Singh & Jaggi, 2020; Ainsworth et al., 2022) of non-transformer architectures do not reveal the full nature of the underlying problem being addressed here.

## 3 BACKGROUND

**Optimal Transport (OT).** OT (Villani et al., 2009) has gained prominence in machine learning for its ability to compare probability distributions effectively, with applications in generative modelling (Arjovsky et al., 2017), class incremental learning (Zhou et al., 2021) and model compression (Li et al., 2021). At its heart, OT aims to find a transport map (TM) $\mathbf{T}$ signifying how much of a discrete source distribution should be moved towards a discrete destination distribution to align the two. This alignment can be hard ($\mathbf{T}$ is a permutation matrix and the solution to the Earth-Mover's Distance, EMD, (Rubner et al., 2000) problem) or can be relaxed yielding a soft alignment (solved

with the Sinkhorn-Knapp algorithm (Knight, 2008)). The softness of the alignment is controlled by a regularization parameter $\lambda_{\text{sinkhorn}}$, where lower values result in harder alignment. More details about OT can be found in the Appendix A.1.

**OTFusion.** Singh & Jaggi (2020) apply this theory to align networks in a layerwise fashion, using either weights or activations as underlying distributions. After the alignment of one or more models to an anchor model, these are then averaged. Formally, for a layer $\ell$ of the model, the transpose of the TM of the previous layer is pre-multiplied with the weight matrix of the current layer: $\widehat{\mathbf{W}}^{(\ell,\ell-1)} \leftarrow \mathbf{T}^{(\ell-1)^\top} \mathbf{W}^{(\ell,\ell-1)}$. The current layer can then be aligned by post-multiplying with the TM of the current layer: $\widetilde{\mathbf{W}}^{(\ell,\ell-1)} \leftarrow \widehat{\mathbf{W}}^{(\ell,\ell-1)} \mathbf{T}^{(\ell)}$. Ainsworth et al. (2022) propose a highly similar approach which, in certain cases, effectively boils down to the same linear programming problem that uncovers (provably and practically) same alignments as OTFusion; thus we continue to base our approach on OTFusion henceforth.

## 4 METHODOLOGY AND IMPLEMENTATION

With a modular architecture like the transformer, it is intuitive to use a divide-and-conquer approach to develop a fusion algorithm. Therefore, we first divide the architecture into its simplest building block — fully connected layers — that can be fused by the prevalent OTFusion strategy. The question remains; how to effectively connect these building blocks, especially if heterogeneous? How to hierarchically reconstruct a fully fused transformer ensuring consistency of the single fused blocks?

As we provide solutions to such open questions, we will guide our discussion in this section with a transport flow perspective, which allows for an intuitive and effective concatenation of blocks of any sort, and that, therefore, in principle can be applied to every architecture. Henceforth, we will use the notation from Vaswani et al. (2017) for Transformers. We display our methods in the non-masked self-attention case, but our method can generalize to the cross-attention or causal masked attention.

### 4.1 TRANSPORTATION MAP FLOW GRAPH

In the typical OTFusion application, the TM of the previous layer is simply passed to the next layer. However, in more complex architectures, the incoming TM of a layer can depend on multiple TMs. To formalize and visualize this flow of TMs, we present the ***Transportation Map Flow Graph***.

To introduce the concept, we use the flow graph of a residual connection (Fig. 1). Rectangles represent the neural network layers; red nodes represent any non-learnable computations or permutations inside the network; edges represent the propagation of the TMs. Layers have exactly one incoming and one outgoing edge. Computation nodes always have multiple incoming edges and one outgoing edge, where the outgoing TM must depend on the incoming TMs. A major contribution of this work is to handle the various complex transportation map flows throughout the transformer architecture.

### 4.2 TRANSFORMER FUSION

#### 4.2.1 RESIDUAL CONNECTIONS

In residual connections, the outputs of a current layer and a residual layer are summed up. The TMs coming from these two layers will be different, therefore the ideal TM flow strategy has to be determined. We explored three heuristics to calculate a weighting vector $\boldsymbol{\gamma}^{(\ell)}$, where each entry $\gamma_i^{(\ell)}$ scales the corresponding rows of the TMs. After obtaining $\boldsymbol{\gamma}^{(\ell)}$ we compute the weighted average as shown in Eq. 1. Find the results in Sec. 5.1.

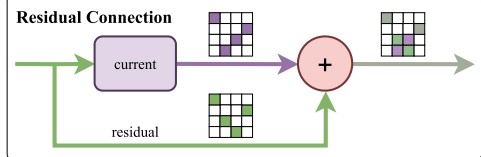

Figure 1: TM flow graph for a residual connection.

$$\mathbf{T}_{\text{out}}^{(\ell)} = \mathbf{T}_{\text{current}}^{(\ell)} \operatorname{diag}(\mathbf{1} - \boldsymbol{\gamma}^{(\ell)}) + \mathbf{T}_{\text{residual}}^{(\ell)} \operatorname{diag}(\boldsymbol{\gamma}^{(\ell)}) \tag{1}$$

**Averaging.** For plain averaging, as proposed by Singh & Jaggi (2020), we set $\forall\, i,\, \gamma_i = 0.5$. This heuristic does not depend on activations and can therefore be used even in the case of weight-based alignment. However, it introduces the strict assumption that the residual and the current layer TM are

of equal importance when aligning the subsequent layer. We therefore extend Singh & Jaggi (2020) with two novel residual policies.

**Weighted Scalar.**   To alleviate the equal contribution constraint from the averaging method, we compute a weighting factor $\forall i, \gamma_i^{(\ell)} = \gamma_{\text{scalar}}^{(\ell)}$ (Eq. 2). We use the activations of the anchor model, over a batch of samples $S$, because only those carry information about the importance of the current and the residual branch in the anchor model to which we try to align the other models. $\mathbf{f}_{\text{residual}}^{(\ell)}(\mathbf{x})$ are the activations from the residual branch while $\mathbf{f}_{\text{current}}^{(\ell)}(\mathbf{x})$ are the activations from the current layer $\ell$.

$$\gamma_{\text{scalar}}^{(\ell)} = \frac{\sum_{\mathbf{x} \in S} ||\mathbf{f}_{\text{residual}}^{(\ell)}(\mathbf{x})||_1}{\sum_{\mathbf{x} \in S} ||\mathbf{f}_{\text{current}}^{(\ell)}(\mathbf{x})||_1 + \sum_{\mathbf{x} \in S} ||\mathbf{f}_{\text{residual}}^{(\ell)}(\mathbf{x})||_1} \tag{2}$$

**Weighted Matrix.**   As opposed to the Weighted Scalar method, here, we calculate a weight vector $\boldsymbol{\gamma}^{(\ell)}$ where each entry $\gamma_i^{(\ell)}$ weighs one strand of a residual connection. The computation of each $\gamma_i^{(l)}$ is similar to Eq. 2 but here we do not compute the $\ell^1$-Norm over the whole activation vectors, instead, we take the absolute value of the corresponding $i$-th values of the activation vectors.

We note that Ainsworth et al. (2022) propose to propagate either the identity ($\mathbf{T}_{\text{out}} = \mathbf{I}$) or the residual transportation map itself ($\forall i, \gamma_i^{(l)} = 1$). In the case of hard alignment, these methods perform worse than averaging.

### 4.2.2   MULTI-HEAD ATTENTION

The attention mechanism (Eq. 3) poses multiple challenges when it comes to TM flow (Fig. 2): what are the incoming TMs for $\mathbf{W}^Q$, $\mathbf{W}^K$ and $\mathbf{W}^V$? Which TM is propagated to $\mathbf{W}^O$? How to handle attention with multiple heads?

$$\text{Self-Attention}(\mathbf{x}) = \text{softmax}(\frac{\mathbf{Q}\mathbf{K}^{\mathbf{T}}}{\sqrt{d_k}})\mathbf{V}, \quad \text{with } \{\mathbf{Q}, \mathbf{K}, \mathbf{V}\} = \mathbf{W}^{\{\mathbf{Q}, \mathbf{K}, \mathbf{V}\}}\mathbf{x} \tag{3}$$

The first challenge is conveniently solved by the TM flow graph. We can simply use the TM from the previous layer for each $\mathbf{W}^Q$, $\mathbf{W}^K$ and $\mathbf{W}^V$. This even holds true for multiple heads. The incoming TM of $\mathbf{W}^O$ is more complex to obtain because it depends on the outgoing TMs of $\mathbf{W}^Q$, $\mathbf{W}^K$, and $\mathbf{W}^V$. However, if we constrain both TMs of $\mathbf{W}^K$ and $\mathbf{W}^Q$ to be equal permutation matrices (i.e., hard alignment with $\mathbf{T}_Q = \mathbf{T}_K = \mathbf{T}_{QK}$), we show that the permutation matrices cancel (see Eq. 4) leaving the softmax undisturbed. Therefore, we only propagate the outgoing TM of $\mathbf{W}^V$ to $\mathbf{W}^O$.

For soft-alignment Eq. 4 no longer holds, in that case we investigated alleviating the constraint of equal TMs for $\mathbf{W}^K$ and $\mathbf{W}^Q$. Removing this constraint slightly increased one-shot accuracy.

$$\widetilde{\mathbf{Q}} = \mathbf{Q}\mathbf{T}_{QK} \quad \text{and} \quad \widetilde{\mathbf{K}} = \mathbf{K}\mathbf{T}_{QK} \quad \text{and} \quad \widetilde{\mathbf{Q}}\widetilde{\mathbf{K}}^\top = \mathbf{Q}\mathbf{T}_{QK}\mathbf{T}_{QK}^\top\mathbf{K}^\top = \mathbf{Q}\mathbf{K}^\top \tag{4}$$

For multi-head attention fusion, there is an additional layer of complexity because one must align the weights and the heads. On top of that, there is no guarantee that a hard one-to-one alignment between heads exists. For that reason, we propose cross-head alignment. During cross-head alignment, $\mathbf{W}_i^Q$, $\mathbf{W}_i^K$ and $\mathbf{W}_i^V$ (where $i$ is the

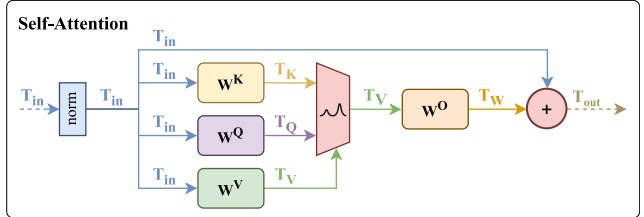

Figure 2: Self-Attention flow graph.

head index) are concatenated across the output dimension to form three combined weight matrices ($\mathbf{W}^Q$, $\mathbf{W}^K$ and $\mathbf{W}^V$). OTFusion is then applied to each of the concatenated weight matrices. Finally, $\mathbf{T}_V$ is propagated to $\mathbf{W}^O$. Find a visualization of our cross-head alignment method in App. B.

### 4.2.3   LAYER NORMALIZATION, EMBEDDINGS AND BIAS

**The layer normalization** is a learnable neural network parameter and consequently must be fused. It contains only two parameters ($\boldsymbol{\alpha}$ and $\boldsymbol{\beta}$) per input and there are no interconnections between different inputs and outputs. Therefore, no TM has to be computed for this layer. The parameters are only aligned w.r.t. to the incoming TM. The incoming TM is then propagated to the subsequent layer.

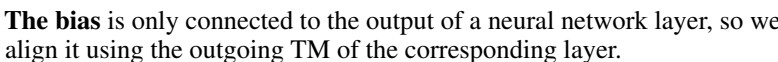

**The ViT embeddings** fusion approach is most effectively conveyed by its TM flow graph, as depicted in Fig. 3. For the concatenation, we notice that the class token is only a small fraction of the full sequence, in other words, for the integrity of the sequence, it is far more important to propagate the TM of the patch embeddings than the one for the class token. After concatenation, the positional embeddings are added. We notice that the addition is the same operation as for residual connections, so we can use one of the three TM flow strategies from Sec. 4.2.1.

Figure 3: ViT embeddings flow graph.

**The bias** is only connected to the output of a neural network layer, so we align it using the outgoing TM of the corresponding layer.

## 4.3 ALIGNMENT STRATEGIES

**Soft vs Hard Alignment.** OTFusion technically allows soft alignment for MLPs, CNNs and ResNets, but Singh & Jaggi (2020) discovered that for these simpler architectures, hard alignment outperforms soft alignment. However, we do not want to limit the search space for optimal alignment to only permutation matrices (possibly too constraining for a complex architecture such Transformers). We, therefore, broaden the perspective on alignment introduced by OTFusion using the Sinkhorn algorithm and tuning the softness of the TM by optimizing over the Sinkhorn regularizer, discovering that soft alignment outperforms hard alignment for Transformers.

**Weights vs. activations alignment.** The combined methodology introduced so far, and the novel perspective on the TM flow, allow us to apply OTFusion to the single fully connected layers without further adaptations in the case of weight-based approach, while the activation-based strategy needs a bit more thought. Transformers operate on sequences of tokens as opposed to simpler architectures that only operate one token at a time. In our activations-based algorithm, we treat every token of the sequence as a possible activation.

**Sequence Filtering.** For ViTs, it is obvious that not every token contributes equally to the final image classification. We hypothesize that activations-based alignment performs best if only the most important tokens of a sequence are considered. Therefore, we explored filtering out unimportant tokens. For datasets where images are centered, we propose window filtering, where only the $n$ by $n$ center patches are considered as activations for activations-based alignment (`window_n`). Additionally, we explored using only the class token for activation-based alignment (`only_cls`).

## 5 EXPERIMENTS AND RESULTS

We evaluate the quality of our approach with two prominent transformer-based architectures: the ViT (Dosovitskiy et al., 2020) and BERT (Devlin et al., 2018). Our focus is to assess the performance and robustness of our proposed fusion techniques in both image and NLP domains. These models offer a direct comparison as they share the same encoder-only architecture. We conducted our experiments on multiple well-known image classification datasets: CIFAR10, CIFAR100, TINY IMAGENET, and IMAGENET-1K. We used Hugging Face both for the implementation of the ViT and for retrieving the datasets. Besides the image classification tasks, we showcase our fusion strategy on the BERT model for an NLP task. We train from scratch multiple BERT models on the masked language modeling (MLM) task over a subset of the Wikipedia dataset, publicly available on the Hugging Face Hub.

**Model Training.** First, we train individual models from scratch on each dataset until *convergence*. We ensure model diversity by initializing each model with different seed values and different batch randomization. This results in unique models with similar performance but located in diverse parts of the landscape, and whose suitable fusion can improve performance. These diverse models, which are rather distant in the parameter space, need a non-trivial alignment strategy to be successfully fused, and therefore exhibit a dramatic drop in performance when fused with a naive approach such as VF. This approximates a plethora of other scenarios (e.g. models trained on different (sub)datasets). Details and training parameters of all models can be found in Appendix C.

**Model Fusion.** We assessed the proposed fusion strategies, and their combination thereof, on the CIFAR10 dataset (refer to the ablation studies in Section 5.1). We measure the performance through the so-called *one-shot* capability, namely the performance of the fused model, without any retraining, on the same task and metric of the parents. This capability is the first important proxy of the capacity of the fusion algorithm to align and then fuse the parent models. The optimal fusion

strategy identified on the CIFAR10 task is then applied to the other tasks and architectures. For each task and alignment strategy (i.e. weights-based and activations-based) we optimize the Sinkhorn regularizer separately (see Fig. 11). The fusion step runs in just seconds on a general-purpose CPU.

**Finetuning.** Besides the *one-shot* performance, similar to Singh & Jaggi (2020); Nguyen et al. (2021), we evaluate the effect of finetuning the fused model. The resulting performance is compared against the single parent models at *convergence* (and thus do not benefit from finetuning), their ensembling, and the VF model that also went through a round of finetuning. Both our fused model and the VF model are optimized separately over a common set of reasonable hyperparameters.

**Note.** We encode the model dimension as (*hidden-layer dimension/intermediate-layer dimension/number of encoders*). Additionally, we report the relative computational burden (latency and FLOPs) below each result table entry.

## 5.1 ONE-SHOT EXPERIMENTS

We optimize the fusion strategy on CIFAR10, searching the configurations previously introduced. In contrast to the observations of Singh & Jaggi (2020) with non-transformer architectures, we observe that a soft-alignment (Sinkhorn) strategy consistently outperforms hard-alignment (EMD). The value of the Sinkhorn regularizer is chosen to maximize the one-shot accuracy (separately for activations- and weights-based alignment). The optimal strategy for handling the residual connections has proven to be the *averaging* policy. Activations-based alignment with the 6x6 window filtering (*window_6*) approach performs best among other filtering strategies and weights-based alignment.

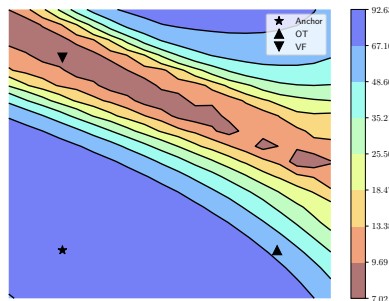

Figure 4: 2D slice of the accuracy landscapes of the anchor and one-shot OT and VF fused models.

In Tab. 1, we present the *one-shot* performance for the best configuration of fusion with the weights-based alignment and the activations-based alignment, both in the scenario with two models and with five models together. VF dramatically drops at random accuracy, while our fusion methodologies are able to preserve most of the capabilities of the individual models. In particular, we achieve the **best accuracy with our soft, activations-based fusion**.

Fig. 4 visualizes a two-dimensional slice of the accuracy landscapes of the anchor model and the two fused models, OT and VF. The visualization is based on the procedure outlined in (Garipov et al., 2018). The plot shows the OT model being in the same basin as the anchor one, while the VF model is separated by a barrier from such basin. This representation effectively underscores the superior performance of our algorithm in comparison to VF, emphasizing its ability to facilitate more dependable knowledge transfer.

Table 1: *One-shot* accuracies on CIFAR10 for the individual parent models, VF, weights-based soft-alignment fusion ($\lambda_{sinkhorn} = 0.06$), activations-based soft alignment ($\lambda_{sinkhorn} = 0.08$) fusion, and activations-based hard-alignment (EMD) fusion. Activations-based is reported with mean and standard deviations over different random seeds. For the best-performing method, we show the absolute increase over VF.

| DATASET | INDIVIDUAL MODELS | VF | OT-WTS (OURS) | OT-ACTS (OURS) | OT-ACTS EMD (OURS) | GAIN OVER VF |
|---|---|---|---|---|---|---|
| CIFAR10 | [92.34, 92.31] | 7.59 | 57.23 | **60.87 ± 0.44** | 24.50 ± 5.66 | +53.28 |
| CIFAR10 | [92.34, 92.31, 92.28, 92.04, 91.47] | 9.47 | 44.46 | **46.56 ± 0.71** | 43.28 ± 2.81 | +37.09 |

**Ablation Studies.** We study the effect of the different OTFusion hyperparameter choices on the *one-shot* performance on the CIFAR10 dataset for two-models fusion. We find that soft alignment (Sinkhorn) outperforms hard alignment (EMD) (see Fig. 5a). We attribute this observation to the flexibility of soft alignment which better accommodates the highly complex nature of the transformer, as multi-head self-attention. We observe a bell-shaped curve with a maximum for a non-zero regularization, thus demonstrating that the optimal alignment is neither hard nor merely soft. We can

therefore optimize this parameter with an inexpensive sweep. Furthermore, as shown in Fig. 5b, the soft alignment for the activations-based fusion is much more stable than hard alignment (EMD) for different seeds of data, suggesting that hard alignment is much more impacted by the activations.

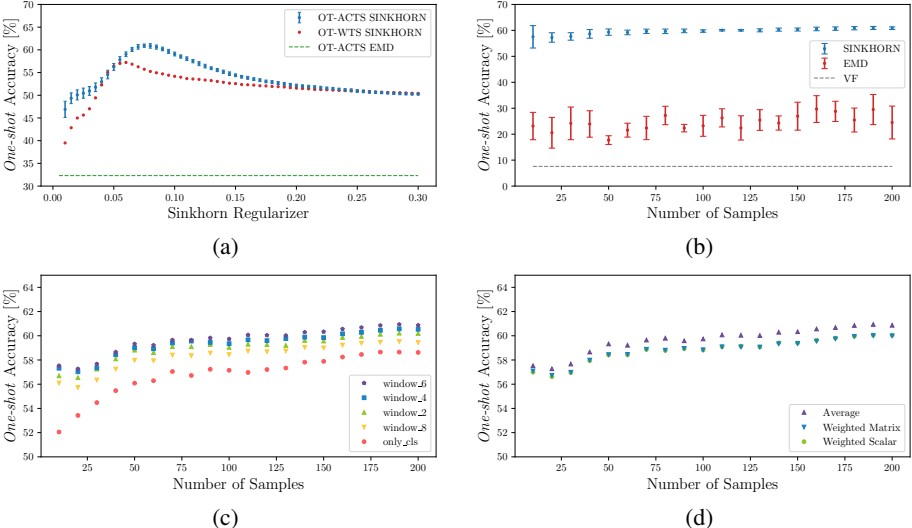

Figure 5: (a) Sinkhorn regularizer effect on *one-shot* performance; (b) stability with different seeds for activations-based fusion over a different number of samples; (c) performance with different activations-filtering strategies for a different number of samples; (d) different transport map policies for residual connections over a different number of samples.

Fig. 5c shows the impact of various filters on the *one-shot* accuracy of the fusion, thereby strengthening our hypothesis that discarding irrelevant activations helps our fusion algorithm converge to a better optimum. Finally, in Fig. 5d we present the impact of the various transport map policies for residuals, as presented in Section 4.2.1. Both weighted policies perform very similarly, slightly falling behind the best accuracy given by the *averaged* policy.

## 5.2 FINETUNED PERFORMANCE

As a last stage of the experimental setup, we finetune the fused models. The performance, as well as the retraining curves, offer an important insight into the quality of the fusion algorithm. While the *one-shot* performance can be heavily impacted by even only a single problematic layer, the capacity of the fused model to effectively, rapidly, and easily recover the performance of the parents allows for a deeper insight into the quality of the fusion across the whole architecture.

Table 2: Post-finetuning accuracies on the CIFAR100 dataset for the individual parent models, their ensemble, VF, weights- and activations-based soft alignment. Model dimension: (384/1536/7).

| DATASET | IND. MODELS | ENS. | FT. VF | FT. OT-WTS | FT. OT-ACTS |
|---------|-------------|------|--------|-----------|-------------|
| CIFAR100 | [64.94, 64.66] ×1 | 68.04 ×2 | 64.91 (-0.03) ×1 | **65.80** (+0.86) ×1 | 65.35 (+0.41) ×1 |
| CIFAR100 | [64.94, 64.66, 64.44, 64.38, 64.34, 64.07] ×1 | 70.71 ×6 | 63.19 (-0.75) ×1 | **65.98** (+1.04) ×1 | 65.25 (+0.31) ×1 |

We show the finetuning results on the widely adopted datasets CIFAR100, and IMAGENET-1K (results on TINY IMAGENET in the Appendix). We first employ our fusion approach on the ViTs trained on the CIFAR100 dataset. As mentioned, we separately optimize the fused model on a common set of hyperparameters, in this case a learning rate (LR) in $\{10^{-3}, 10^{-4}, 10^{-5}\}$ and the number of epochs in $\{10, 20, 100, 200\}$. In Tab. 2 we observe that **both our soft-alignment strategies** (i.e. with weights- and activations-based alignment) **are capable of outperforming the converged parents**, with the gain that increases with the number of parent models. This suggests a successful knowledge transfer of the parents into the fused model. While the obtained accuracy lacks

behind the ensembling performance, in our scenario there is no computational overhead, while the cost of the ensembling model grows linearly with the number of models.

Table 3: Accuracies on the IMAGENET-1K dataset after finetuning for the individual parent models, their ensemble, VF, and weights-based soft alignment. Model dimension: (384/1536/12).

| DATASET | IND. MODELS | ENS. | FT. VF | FT. OT-WTS |
|---|---|---|---|---|
| IMAGENET-1K | [75.33, 74.88] | 76.56 | 67.83 (-7.50) | **75.80** (+0.47) |
| | ×1 | ×2 | ×1 | ×1 |

In Tab. 3 we present further results on the challenging and widely-adopted IMAGENET-1K dataset. The results are consistent with those found in the CIFAR100 case, strengthening the *general applicability* of our methods, and its *scalability to larger models and more challenging datasets*. We also stress the fact that, especially with this difficult dataset, even after finetuning, VF fails to recover a comparable accuracy, converging to suboptimal performance.

In this work, we focused on the vision application of the Transformer architecture, but our method is agile to architectural changes, and we demonstrate its wide applicability to the BERT model. Although preliminary explorations of our fusion strategy on the BERT model show some differences with respect to the ViT case (more details on this in App D), the results are on par with those presented above. In particular, the fused and finetuned model, outperforms both parents and VF on the widely adopted *GLUE* benchmark (Wang et al., 2018). The results are presented in Tab. 17 of the App. E.

We want to highlight an insight into the finetuning process. In particular, we have observed that the *best accuracy of our fused models is achieved extremely quickly*, as much as two orders of magnitude fewer steps needed to train the parents from scratch, and, as a comparison, VF requires far higher computation to reach a comparable (but worse) performance. For further exemplification refer to Fig. 12 in Appendix E.2.

Our methodology, as opposed to VF, works out of the box with models having different widths (heterogeneous fusion). *We find a consistent absolute increase in test accuracy over the performance of the smaller anchor network*, thus implying successful knowledge transfer (Tab. 4). These results showcase that our method is an effective and *efficient alternative to knowledge distillation*.

Table 4: Results for heterogeneous fusion on CIFAR100. VF cannot be applied here.

| ANCHOR | LARGER | ENS. | FT. OT-WTS |
|---|---|---|---|
| 63.18 | 64.94 | 67.66 | **64.11** (+0.93) |
| ×1 | ×4 | ×5 | ×1 |
| (192/768/7) | (384/1536/7) | | (192/768/7) |
| 64.07 | 64.79 | 67.94 | **64.88** (+0.81) |
| ×1 | ×2.3 | ×3.3 | ×1 |
| (384/1536/7) | (576/2304/7) | | (384/1536/7) |

## 6 DISCUSSION

The fusion methodology for transformer models proposed in this paper is easily adapted to different architectural variants and is readily applicable to models of different widths. However, heterogeneous fusion of networks of different depths is a common limitation of the predominant fusion methods (Singh & Jaggi, 2020; Ainsworth et al., 2022) which are inherently based on a sequential layerwise alignment. Consequently, we too inherit a similar limitation when expanding fusion to the case of Transformers. Overall, this is undoubtedly a fascinating research challenge to extend Transformer fusion (or, broadly speaking, fusion at large) to heterogeneous depth settings which, however, is outside the scope of the current work.

**In summary**, we showcased how distinct independently trained transformer networks can be combined through the lens of Optimal Transport. Utilizing a novel graph interpretation of the transportation map flow, we developed an algorithm for fusing multiple transformer networks that extends the existing fusion techniques and that specifically caters to the idiosyncrasies of the transformer architecture. We also uncovered an intriguing benefit of using soft alignment when fusing Transformers, which had been under-utilized in the past. Overall, we showed that our technique can retain most of the performance of the converged parent models in *one-shot*, and even outperforms them after finetuning, across multiple vision and NLP tasks proving the scalability and wide applicability of our methods thereby providing a highly efficient and promising alternative to ensembling. Finally, our algorithm successfully applies to the fusion of models of different sizes, too, efficiently transferring knowledge from larger to smaller Transformers, and thus offering an effective alternative to distillation.

## ACKNOWLEDGEMENTS

Sidak Pal Singh would like to acknowledge the financial support from Max Planck ETH Center for Learning Systems.

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

## A  Background on Optimal Transport and OTFusion

### A.1  Optimal Transport Theory

At its core, Optimal transport (OT) provides a way to compare two (or more) probability distributions $\mu := (\mathbf{a}, \mathbf{X}) = \sum_{i=1}^{n} a_i \cdot \delta(\mathbf{x}_i)$ and $\nu := (\mathbf{b}, \mathbf{Y}) = \sum_{j=1}^{m} b_j \cdot \delta(\mathbf{y}_j)$, where $\delta(\cdot)$ is the Dirac-delta. These distributions are typically supported in a high-dimensional space, i.e., $\mathbf{x}_i \in \mathcal{X} = \mathbb{R}^{d_1}$, and $\mathbf{y}_j \in \mathcal{Y} = \mathbb{R}^{d_2}$, $\forall i, j$, and also where, being distributions, $\sum_{i=1}^{n} a_i = \sum_{j=1}^{m} b_j = 1$. These given distributions, in our case, may correspond to neurons or weights in a particular layer of the two networks. OT aims to find a transport plan $\mathbf{T}$ (or map) that signifies how much of these weights of the source model, should be moved towards the destination model, while adhering to the geometry of the underlying 'ground' space, usually available in the form of a 'ground metric', e.g., $\mathbf{C}_G(\mathbf{x}, \mathbf{y}) = \|\mathbf{x} - \mathbf{y}\|_2^2$ in the Euclidean case. Mathematically, one can formulate OT through an equivalent linear program:

$$\mathrm{OT}(\mu, \nu; \mathbf{C}) := \min \langle \mathbf{T}, \mathbf{C} \rangle_F \quad \text{s.t.,} \quad \mathbf{T}\mathbb{1}_m = \mathbf{a}, \mathbf{T}^\top \mathbb{1}_n = \mathbf{b} \quad \text{and} \quad \mathbf{T} \in \mathbb{R}_+^{(n \times m)}.$$

where appropriate mass conservation and positivity constraints are met. Here, $\langle \cdot, \cdot \rangle_F$ is the Frobenius inner product and $\mathbb{1}_n \in \mathbb{R}^n$ denotes a vector containing all ones of size $n$. While the above problem will find a solution at the vertex of the polytope, one can relax the search to smooth solutions by regularizing the entropy $h$ of the transport plan (Cuturi, 2013), i.e., $h(\mathbf{T}) = \sum_{i,j} -T_{ij} \log(T_{ij})$

$$\mathrm{OT}_\lambda(\mu, \nu; \mathbf{C}) := \min \langle \mathbf{T}, \mathbf{C} \rangle_F - \lambda \, h(\mathbf{T}) \quad \text{s.t.,} \quad \mathbf{T}\mathbb{1}_m = \mathbf{a}, \mathbf{T}^\top \mathbb{1}_n = \mathbf{b} \quad \text{and} \quad \mathbf{T} \in \mathbb{R}_+^{(n \times m)}.$$

Besides allowing for a soft assignment, it also allows for an efficient solution via the Sinkhorn-Knapp algorithm (Knight, 2008) that results in a speed-up by an order of magnitude in the dimension $d_1$ (or $d_2$) and can be parallelized on GPUs. In contrast, the unregularized problem, which is also commonly referred to as the Earth-Mover's Distance (EMD; Rubner et al. (2000)), scales cubically in the dimension.

### A.2  OTFusion

OTFusion (Singh & Jaggi, 2020) first aligns several models: $B, C, \ldots$, to an anchor model $A$. Then, the aligned models are averaged. Alignment is implemented through transportation maps, obtained by calculating the minimal transport cost between activations or weights of the neurons that should be aligned, giving rise to two different approaches, namely activations- and weights-based respectively. The OTFusion process works in a sequential fashion; assuming models with a specific depth $L$, each of the models' layers, at layer $\ell$, are aligned before moving to the next layer $\ell + 1$. First, the transpose of the transportation map of the previous layer is pre-multiplied with the weight matrix of the current layer: $\widehat{\mathbf{W}}_B^{(l,l\text{-}1)} \leftarrow \mathbf{T}^{(l\text{-}1)^\top} \mathbf{W}_B^{(l,l\text{-}1)}$. The current layer can then be aligned by post-multiplying with the transportation map of the current layer: $\widetilde{\mathbf{W}}_B^{(l,l\text{-}1)} \leftarrow \widehat{\mathbf{W}}_B^{(l,l\text{-}1)} \mathbf{T}^{(l)}$.

## B  Cross-Head Alignment Visualisation

Fig. 6 visualizes the cross-head alignment algorithm for a tiny multi-head self-attention block. The aligned weights can then be averaged with the corresponding weights of the anchor model to get the weights for the OTFused model.

## C  Experimental Setup

### C.1  Vision Transformer - *CIFAR10*, *CIFAR100*, *Tiny ImageNet* and *ImageNet-1k*

**Model Details**  We use the ViT implementation available on Hugging Face[3] and we train it from scratch, without using any pre-trained weights. The architectural details of the model can be seen in Table 5.

---

[3] https://huggingface.co/docs/transformers/model_doc/vit

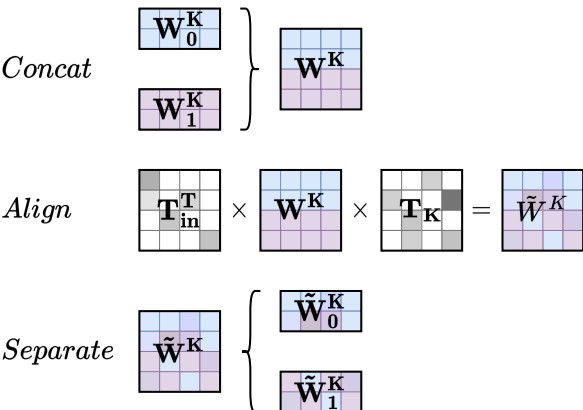

Figure 6: Visualization of the cross-head alignment algorithm for a multi-head attention block with $h = 2$, $d_{head} = 2$, $d_{model} = 4$, where $h$ is the number of heads, $d_{head}$ is the head dimension and $d_{model}$ is the model dimension.

Table 5: Parameters for the ViT models.

| | | |
|---|---|---|
| Input image size | *CIFAR10/100* | 32x32x3 |
| | *Tiny ImageNet* | 64x64x3 |
| Patch extraction | | Convolutional |
| Patch dimension | | 4x4 |
| Number of layers | | 7 |
| Number of heads | | 12 |
| Size of embeddings | | 384 |
| Intermediate size | | 1536 |
| Non-linearity | | GELU |

**Image Augmentation**    We applied two different image augmentation policies on the *CIFAR 10/100* and *Tiny ImageNet* datasets to achieve satisfactory training performance. For the CIFAR datasets, the augmentations have been adapted from an open-source implementation[4], while for *Tiny ImageNet* the Autoaugment[5] class from Pytorch has been used.

**Training Details**    Training details are reported in Table 6. Figures 7, 8, 9 show the training curves for the *CIFAR10*, *CIFAR100*, and *Tiny ImageNet* respectively.

Table 6: Training details for the ViT models trained on CIFAR and *Tiny ImageNet* models.

| Optimizer | | AdamW |
|---|---|---|
| Weight decay | | $5 \cdot 10^{-5}$ |
| Learning Rate | | Maximum value of $1 \cdot 10^{-3}$ |
| LR Scheduler | | Cosine scheduling |
| Warmup | | 0.025% epochs of warmup |
| Training Epochs | *CIFAR* | 2500 |
| | *Tiny ImageNet* | 250 |
| Batch size | *CIFAR* | 1024 |
| | *Tiny ImageNet* | 256 |
| Gradient accumulation | *CIFAR* | 2 |
| | *Tiny ImageNet* | 8 |
| Random seed | | 0-4 |

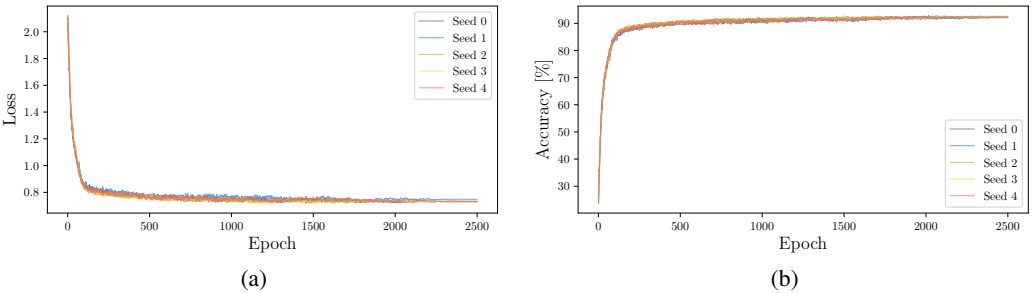

(a)            (b)

Figure 7: Training curves for the *CIFAR10* dataset over five different seeds. (a) Validation loss; (b) validation accuracy.

## C.2    VISION TRANSFORMER - IMAGENET

**Model Details**    We use the *SimpleViT* class from vit-pytorch[6] and we train it from scratch, without using any pre-trained weights. The architectural details of the model can be seen in Table 7.

---

[4]https://github.com/DeepVoltaire/AutoAugment
[5]https://pytorch.org/vision/main/generated/torchvision.transforms.AutoAugment.html
[6]https://github.com/lucidrains/vit-pytorch

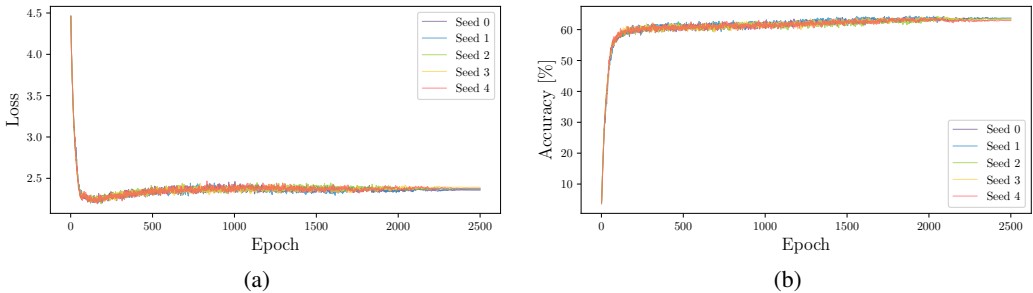

Figure 8: Training curves for the *CIFAR100* dataset over five different seeds. (a) validation loss; (b) validation accuracy.

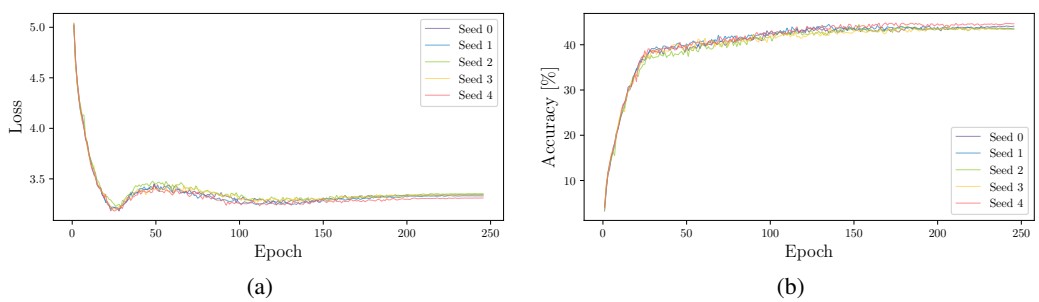

Figure 9: Training curves for the *Tiny ImageNet* dataset over five different seeds. (a) validation loss; (b) validation accuracy.

Table 7: Parameters for the ViT models.

| | |
|---|---|
| Input image size | 224x224x3 |
| Patch extraction | Linear |
| Patch dimension | 16x16 |
| Number of layers | 12 |
| Number of heads | 6 |
| Size of embeddings | 384 |
| Intermediate size | 1536 |
| Non-linearity | GELU |

**Image Augmentation**   We first applied *RandomResizedCrop()* and *RandomHorizontalFlip()* to the input image form Pytorch transforms sub-package [7]. Then we applied the Autoaugment class from the same Pytorch sub-package. Images are then normalized with $\mu = [0.485, 0.456, 0.406]$ and $\sigma = [0.229, 0.224, 0.225]$.

**Training Details**   Training details are reported in Table 8.

Table 8: Training details for the ViT models trained on Imagenet.

| Optimizer | AdamW |
|---|---|
| Weight decay | $1 \cdot 10^{-4}$ |
| Learning Rate | Maximum value of $1 \cdot 10^{-3}$ |
| LR Scheduler | Cosine scheduling |
| Training Epochs | 90 |
| Batch size | 1000 |
| Random seed | 2,4 |

### C.3   PROFILING INFORMATION

In Tab. 9 we provide profiling information for our most used ViT configuration.

Table 9: Profiling information for our most used ViT configuration. The experiments were run on an RTX 4090. We count one fused-multiply accumulate instructions as one FLOP. Different datasets have different image resolutions, leading to different sequence lengths propagating through the transformer, which affects the computational expense of a forward pass.

| MODEL MODEL DIM. | #PARAMS (M) | DATASET | #PATCHES | FLOPS (B) | TP (IMAGE/S) |
|---|---|---|---|---|---|
| VIT | 12.4 | *CIFAR100* | 65 | 0.8 | 13.2 K |
| (384/1536/7) | | *Tiny ImageNet* | 257 | 3.5 | 2.4 K |

### C.4   BERT

**Model Details**   We use the BERT implementation available on Hugging Face[8] together with the pre-trained `bert-base-uncased` tokenizer [9]. Our BERT model has the architectural details presented in Tab. 10.

**Training Details**   We train the BERT models, from scratch, over five different seeds. Training details are shown in Tab. 11.

We use a MLM task on a subset of the Wikipedia dataset, available on Hugging Face [10], with an MLM probability of 0.15.

The training curve of the loss, for one seed, is presented in Fig. 10.

---

[7] https://pytorch.org/vision/stable/transforms.html
[8] https://huggingface.co/docs/transformers/model_doc/bert
[9] https://huggingface.co/docs/transformers/main_classes/tokenizer
[10] https://huggingface.co/datasets/wikipedia/viewer/20220301.simple

Table 10: Parameters of the architecture for the BERT models.

| | |
|---|---|
| Number of encoders | 6 |
| Number of heads | 12 |
| Size of embeddings | 768 |
| Intermediate size | 3072 |
| Maximum position embedding | 512 |
| Attention dropout probability | 0.1 |
| Hidden dropout probability | 0.1 |
| Non-linearity | GELU |

Table 11: Training details for the BERT models.

| | |
|---|---|
| Optimizer | `AdamW` |
| Learning Rate | cosine scheduling with 4 epochs of warmup; maximum value of $5 \cdot 10^{-5}$ |
| Training Epochs | 40 |
| Batch size | 16 |
| Random seed(s) | 0-4 |

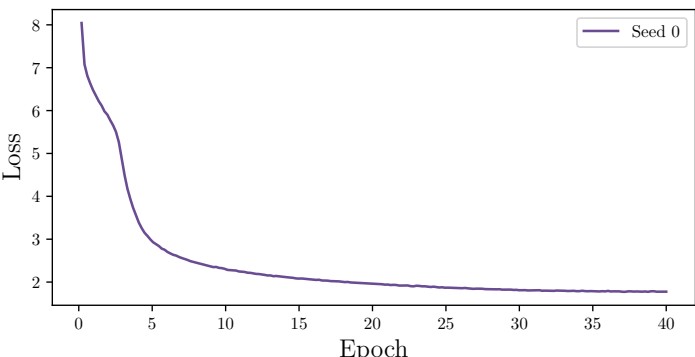

Figure 10: BERT pre-training validation loss for random seed 0.

# D    SINKHORN REGULARIZER ABLATIONS

The Sinkhorn algorithm, and in general the soft alignment paradigm, has been heavily underused in literature and therefore there is little information about its impact on OTFusion. As presented above, we uncover intriguing behaviors, that require reconsidering its use. In the following Sections, we extend our findings related to soft alignment, in particular with the role of the regularization parameter.

## D.1    ABLATION ON RESNET

To compare the findings for the transformer architecture, we also investigate the effect of the Sinkhorn regularizer on the ResNet architecture (Fig. 11a). In agreement with the findings of Singh & Jaggi (2020), the best result is achieved with EMD, and a small regularizer is preferred as it approaches the hard alignment. This result is thus suggesting an opposite behavior when it comes to soft alignment since the transformer benefits from a soft alignment.

## D.2    ABLATIONS ON *CIFAR100*, *Tiny ImageNet*, BERT MLM TASK

In Fig. 11 we present the effect of the Sinkhorn regularizer on the other considered datasets, namely *CIFAR100* (Fig. 11b) and *Tiny ImageNet* (Fig. 11c) for the ViT, and the MLM task on the Wikipedia subset, for BERT (Fig. 11d).

The outcomes for *CIFAR100* and *Tiny ImageNet* are in line with the results of the *CIFAR10* case, namely a non-zero regularizer achieves the optimal performance.

As hinted in Sec. 5.2, we have observed some differences in the regularization effect on the BERT model. This difference can be observed in Fig. 11d, where we plot the effect of the regularization parameter on the validation loss. We observe that, in contrast to the observations for the ViT, the loss curve shows no inverted bell curve, suggesting that there is no finite optimal regularizer, i.e. that a completely soft alignment is best suited for this model.

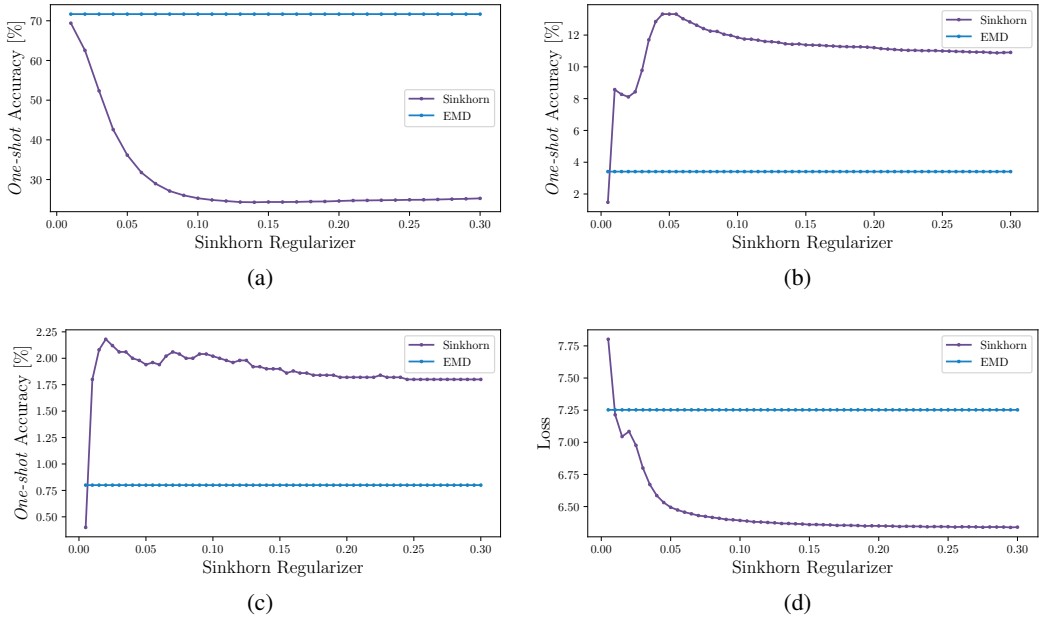

Figure 11: Sinkhorn regularizer effect on *one-shot* performance. EMD-fusion performance is shown as a reference. (a) Accuracy for *ResNet* on *CIFAR10* (higher is better); (b) accuracy for ViT on *CIFAR100* (higher is better); (c) accuracy for ViT on *Tiny ImageNet* (higher is better); (d) loss for BERT on MLM task (lower is better).

### D.3 What Happens at the Extreme Edge of Sinkhorn Regularization?

As presented above, the softness of the alignment is impacted by the Sinkhorn regularizer. If the regularizer is close to zero, the algorithm converges to a permutation matrix (i.e. hard alignment); in contrast, if the regularizer is very large, the algorithm converges to a unit-matrix divided by the dimension of itself.

#### D.3.1 Sinkhorn Regularizer to Zero

In general, we have observed that the smaller the regularizer becomes, the harder the alignment gets. However, for very small Sinkhorn regularizer values the algorithm breaks down. This is especially visible in Fig. 11b and 11c where for the smallest regularizer the *one-shot* accuracy falls below the *one-shot* accuracy of EMD. We found that normalizing the cost matrix and the activations/weights to calculate the cost matrix, pushes the breakdown closer to zero and thus improving stability.

#### D.3.2 Sinkhorn Regularizer to Infinity

We conducted an experiment to show that even in the case of extreme regularization (i.e. completely soft alignment) information is transferred from model B to the anchor model. In this experiment, we fuse a randomly initialized model (10% accuracy on *CIFAR10*) with a model at convergence (92% accuracy on *CIFAR10*). The *one-shot* accuracy for this experiment is 10%. On the other hand, if we fuse two converged models, we get a *one-shot* accuracy of 47% for a completely soft alignment. This suggests that, even in the highly regularized case, our algorithm allows knowledge transfer.

# E    FURTHER RESULTS

In this section, we provide more results from our experiments. We report both *one-shot* and finetuned accuracies over the datasets of choice.

## E.1    *One-shot*

Tab. 12 and Tab. 13 report the *one-shot* accuracies for *Tiny ImageNet* and *CIFAR100* datasets, respectively.

Table 12: *One-shot* accuracies on the *Tiny ImageNet* dataset for the individual parent models, their ensemble, VF, weights-based soft-alignment fusion, and activations-based soft alignment fusion. The last column shows the highest finetuned performance as a comparison. Activations-based is reported with mean and standard deviations over different data seeds. The figure beneath the test accuracies signifies how much more computation is required by the model ensemble with respect to our fusion technique.

| DATASET | INDIVIDUAL MODELS | ENS. | VF | OT-WTS (OURS) | OT-ACTS (OURS) | FT. OT-WTS (OURS) |
|---------|-------------------|------|-----|---------------|----------------|-------------------|
| *Tiny ImageNet* | [45.30, 45.22, 44.50, 44.36, 43.78] | 51.28 $\times 5$ | 0.44 $\times 1$ | 1.64 $\times 1$ | $3.03 \pm 0.27$ $\times 1$ | **45.90** x1 |

Table 13: *One-shot* accuracies on the *CIFAR100* dataset for the individual parent models, their ensemble, VF, weights-based soft-alignment fusion, and activations-based soft alignment fusion. The last column shows the highest finetuned performance as a comparison. Activations-based is reported with mean and standard deviations over different data seeds. The figure beneath the test accuracies signifies how much more computation is required by the model ensemble with respect to our fusion technique.

| DATASET | INDIVIDUAL MODELS | ENS. | VF | OT-WTS (OURS) | OT-ACTS (OURS) | FT. OT-WTS (OURS) |
|---------|-------------------|------|-----|---------------|----------------|-------------------|
| *CIFAR100* | [64.94, 64.66] | 68.04 $\times 2$ | 0.77 $\times 1$ | 13.32 | $11.70 \pm 0.13$ $\times 1$ | **65.80** |
| *CIFAR100* | [64.94, 64.66, 64.44, 64.38, 64.34, 64.07] | 70.71 $\times 6$ | 0.98 $\times 1$ | 11.16 | $7.45 \pm 0.25$ $\times 1$ | **65.98** |

## E.2    FINETUNING

After fusing the models, we finetune them. Finetuning parameters and results are reported in the subsections below.

### E.2.1    FINETUNING DETAILS - VIT

As mentioned in Sec. 5, we finetune VF and our fused models separately on a common set of hyperparameters. In the following paragraph the subset used over the different datasets and models:

- ViT - *CIFAR100*: LR in $\{10^{-3}, 10^{-4}, 10^{-5}\}$, number of epochs in $\{10, 20, 100, 200\}$
- ViT - *Tiny ImageNet*: LR in $\{10^{-3}, 10^{-4}, 10^{-5}\}$, number of epochs in $\{1, 2, 10, 20\}$

Finetuning on the *ImageNet-1k* dataset is inherently expensive. We have thus finetuned for just 8 to 10 epochs the fused models, with an LR of $10^{-4}$. The boost in performance presented in Tab. 2 is thus even more noteworthy given the limited capacity to exhaustively find suitable hyper-parameters for finetuning.

### E.2.2    RESULTS

**Vision Transformer**    In Tab. 14 we report the finetuning results for the fusion and ensemble of two and six models on the *CIFAR100* dataset. The results show how weight-based soft alignment outperforms both weight-based hard alignment and activation-based soft alignment. Furthermore, in Tab. 15 we present further results on the *Tiny ImageNet* dataset.

Table 14: Accuracies on the *CIFAR100* dataset after finetuning for the individual parent models, their ensemble, VF, weights-based soft alignment, weight-based hard alignment, and activations-based soft-alignment. The figure beneath the test accuracies signifies how much more computation is required by the model ensemble with respect to our fusion technique.

| DATASET | INDIVIDUAL MODELS | ENS. | FT. VANILLA | FT. OT-WTS (OURS) | FT. OT-WTS EMD (OURS) | FT. OT-ACTS (OURS) |
|---|---|---|---|---|---|---|
| *CIFAR100* | [64.94, 64.66] | 68.04 | 64.91 | **65.80** | 64.72 | 65.35 |
| | | ×2 | ×1 | ×1 | ×1 | ×1 |
| *CIFAR100* | [64.94, 64.66, 64.44, 64.38, 64.34, 64.07] | 70.71 | 63.19 | **65.98** | 65.42 | 65.25 |
| | | ×6 | ×1 | ×1 | ×1 | ×1 |

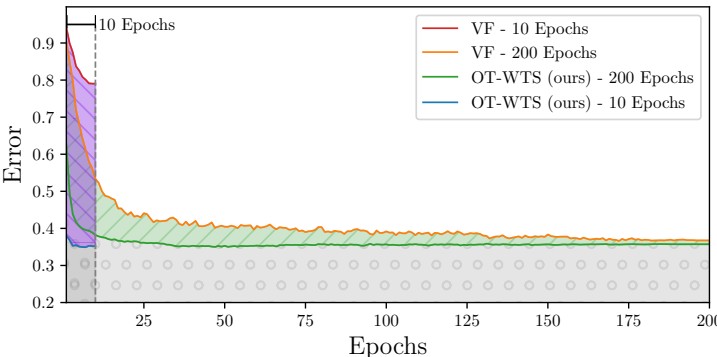

Figure 12: Finetuning curves on the validation set. Cosine scheduling is used. Validation error on the CIFAR100 dataset.

**BERT**    The results after finetuning for the BERT model are presented in Tab. 16 and Tab 17.

Table 15: Accuracies on the *Tiny ImageNet* dataset after finetuning for the individual parent models, their ensemble, VF, weights-based soft alignment, and activations-based soft alignment. Model dimension is encoded as (*hidden-layer dimension/intermediate-layer dimension/number of encoders*). The figure beneath the accuracies indicates the relative computational burden (latency and FLOPs) of the model(s).

| DATASET | IND. MODELS | DIMENSION | ENS. | FT. VF | FT. OT-WTS | FT. OT-ACTS |
|---------|-------------|-----------|------|--------|------------|-------------|
| *Tiny ImageNet* | [45.30, 45.22, 44.50, 44.36, 43.78] | (384/1536/7) | 51.28 | 38.82 | 45.44 | **45.90** |
| | $\times 1$ | | $\times 5$ | $\times 1$ | $\times 1$ | $\times 1$ |

Table 16: Loss values for BERT on the MLM task after finetuning for the individual parent models, their ensemble, VF, and weights-based alignment fusion. Both VF and our fused model are trained with a LR of $5 \cdot 10^{-5}$ for only 2 epochs. This shows the much faster speed of recovery of our approach, compared to VF. The figure beneath the test accuracies signifies how much more computation is required by the model ensemble with respect to our fusion technique.

| DATASET | INDIVIDUAL MODELS | ENS. | FT. VANILLA | FT. OT-WTS (OURS) |
|---------|-------------------|------|-------------|-------------------|
| MASKED WIKI | [1.612, 1.761, 1.776, 1.794, 1.807] | 1.665 $\times 5$ | 2.946 $\times 1$ | 2.224 $\times 1$ |

Table 17: Results for BERT evaluation on GLUE benchmark, after finetuning for 14 epochs. Accuracy is the metric for SST2, QNLI, RTE and WNLI. Matthews corr. is the metric for COLA. F1/Accuracy is the metric for MRPC and QQP. Pearson/Spearman corr. is the metric for STSB. Matched acc./Mismatched acc. is the metric for MNLI.

| TASK | PARENT | OT | VF |
|------|--------|-----|-----|
| MRPC | 0.852/**78.2** | **0.853**/77.7 | 0.807/72.1 |
| STSB | 0.828/0.827 | **0.841/0.838** | 0.771/0.771 |
| QQP | 0.844/88.2 | **0.847/88.5** | 0.840/88.1 |
| MNLI | **76.1/76.4** | 75.9/76.1 | 74.1/74.6 |
| COLA | 0.263 | **0.275** | 0.236 |
| QNLI | 84.1 | **85.1** | 83.0 |
| WNLI | 26.8 | **29.4** | 27.6 |
| SST2 | 85.6 | **86.5** | 84.9 |
| RTE | 62.1 | **63.4** | 51.6 |

