# OpenReview forum: "Transformer Fusion with Optimal Transport"
_ICLR.cc/2024/Conference — ICLR 2024 poster_

### Official Review · Reviewer_hByk · 2023-10-31

**Soundness:** 3 good
**Presentation:** 4 excellent
**Contribution:** 2 fair
**Rating:** 8
**Confidence:** 3

**Summary:**

The authors propose a method for fusing multiple independently trained transformer architectures using optimal transport to align their respective architectural components. To this end, authors analyse predominant Transformer architectures based on their components, and provide OTFusion methods for each. The proposed approach allows for fusing transformers of different sizes.
Experiments are conducted on a range of image classification datasets; CIFAR10, CIFAR100, TinyImagenet and ImageNet-1k. Authors show results for models obtained through both zero-shot (without fine tuning) and with fine tuning. In zero-shot fusion, the proposed approach outperforms Vanilla Fusion methods. With fine tuning, the proposed approach is able to beat either parent model. Authors conclude with a number of limitations of their approach and suggestions for future research.

**Strengths:**

This paper is very well-written, authors give clear and concise descriptions of their approach and illustrate its complex aspects through figures and examples. This makes the manuscript easy to read and the ideas it expands upon easy to understand despite their complexity. The method seems to work well, drastically outperforming Vanilla Fusion (naive model averaging). The authors show a range of valuable ablations, and motivate most of their design choices well.

**Weaknesses:**

My main concern is to do with the clarity of the contribution of this work. The authors refer to [1] a lot in their paper, where the concept of OTFusion is introduced. It seems like a lot of the techniques used in this work were actually introduced there. Although I understand the need for reintroducing these concepts in the manuscript for contextual clarity, I think it would be good to give a clearer picture of the actual contributions made in this work and the methods proposed in previous works. From the description under 4.3 it seems [1] uses hard alignment where you find soft alignments to outperform. Are these contributions of your work? What about the TM combination approaches (Averaging/Weighted Scalar/ Weighted Matrix)? Or heterogeneous fusion?

I hope the authors are able to address this in their rebuttal, in which case I see this work as an interesting and strong submission.

[1] Sidak Pal Singh and Martin Jaggi. Model fusion via optimal transport. Advances in Neural Information
Processing Systems, 33:22045–22055, 2020.

**Questions:**

-What do you mean by “This diversity offers a challenging fusion problem requiring a non-trivial alignment strategy, and thus effectively recreates a plethora of other scenarios” (under 5 - Model Training). Can you explain e.g. how varying random seed equates to model training on different subsets?
-How does your work relate to [2]? You indicate that [2] is very similar to OTFusion, but looking at zero-shot performance of your method (and your VF baseline) on CIFAR10 classification it seems performance is drastically different (~93% vs ~60%). If essentially identical, why does [2] yield zero-barrier LMC where your approach does not?
-Could you give an intuition for soft-alignment, what resulting network is actually being constructed  in this case and why could it be beneficial compared to hard alignment approaches?
-Do you have an intuition for why your method performs better with soft-alignment, where [1] shows better results with hard alignment?

[2] Samuel K Ainsworth, Jonathan Hayase, and Siddhartha Srinivasa. Git re-basin: Merging models
modulo permutation symmetries. arXiv preprint arXiv:2209.04836, 2022.

---

Update after rebuttal: I thank the authors for their thorough rebuttal. I'm pleased to say my concerns are adequately addressed. Also considering the largely positive reviews by the other reviewers, I'd like to update my recommendation to an accept.

---

> ### Author Response · Authors · 2023-11-17
> **Rebuttal by Authors 1/3**
>
> *Thank you for your useful feedback. We are truly glad that you have appreciated our efforts for a well-presented and well-written paper. At the same time, we recognize that some explicit clarifications regarding our contribution would have helped to better convey the contribution of our paper, and we are grateful for making us aware of this. To reflect your doubts, and with the utmost goal of improving the clarity of our work, we have implemented various changes throughout the manuscript.*
>
> ---
>
> In what follows, besides addressing your specific questions, we would nevertheless like to first expand on your concerns about the contribution of our work.
>
> ### Contribution of this work with respect to [1]
>
> **Key differences**
>
> 1. **Enabling Fusion for newer Transformer-based architectures**
>
>     [1] is the first to successfully introduce the concept of Optimal Transport for alignment and fusion of multiple models. But, broadly speaking, i**t is however restricted to simple architectures** such as MLPs, CNNs, and instances of ResNet.
>
>     It is **not** equipped in any way to align and fuse models with complex information streams and to fuse transformer-specific components such as multi-head attention layers, layer-normalization, embeddings, or the sequential nature of the data, which we systematically analyze and successfully handle.
>
> 2. **New findings in regards to hard vs soft alignment**
>
>     Furthermore, we broaden the perspective on alignment introduced by [1] in the following manner. While [1] technically allows soft alignment, they discovered that for simpler architectures (MLPs, CNNs, ResNets) hard alignment outperforms soft alignment (Table S4 of [1]).
>
>     In contrast, **we find the reverse to be true when fusing Transformers** (Sec. 4.3, and Sec. 5), which possibly hints at the difference in the nature of their architectures and representations, and forms an interesting question for standalone future study.
>
> **Other aspects**
>
> - **Heterogeneous fusion**
>
>     As mentioned on page 2 of [1], indeed [1] is the first to showcase heterogeneous model fusion, “OTFusion accommodates the fusion of models with different widths, and in turn, different sizes”.  But, **we are the first to extend this concept to Transformers**.
>
> - **Weighted TM combination & other minor methodological extensions:**
>
>     Lastly, we have included some other extensions for OTFusion [1], to better support transformers.
>
>     As an example, realizing the potentially complex peculiarities of transformers when it comes to residual streams, we have extended the Averaging strategy introduced by [1], with **two novel strategies**, namely *Weighted Scalar* and *Weighted Matrix*, that allow for a more flexible fusion strategy depending on the properties of the residual stream of that layer (Sec. 4.2.1).
>
>     Also, we have extended OTFusion's idea of activations-based alignment to better encompass the sequential nature of the data, introducing various **novel strategies** of activations filtering in the case of the ViT (Sec. 4.3).
>
> ---

---

> > ### Author Response · Authors · 2023-11-17
> > **Rebuttal by Authors 2/3**
> >
> > ### Random seed vs. subset training
> >
> > Varying random seeds leads to optimizing in altogether different regions of the landscape. In contrast, if the networks are initialized from the same point in the landscape, but trained on different subsets, the ensuing optimization trajectories are different but nevertheless reside in more or less the same region of the landscape. The latter idea is supported by [5], and [6], who mention that networks trained “from the same initialization on disjoint subsets” are “connected by linear paths of constant test error“.
> >
> > So here by “diversity”, we mean it more in the sense of networks being located in **diverse parts of the landscape**, and whose suitable fusion can improve performance. These diverse models, which are rather distant in the parameter space, need a non-trivial alignment strategy to be successfully fused as presented here.
> >
> > We have adjusted our formulation in the paper to make this more clear.
> >
> > ---
> >
> > ### Relation of our work to [2]
> >
> > Firstly, [2] proposes an approach highly similar to OTFusion [1] and, in certain cases (such as when using activation-based alignment), it even uncovers the same alignment as OTFusion.
> >
> > Next, to answer your question, we explain why the apparent difference in the performance on CIFAR10 relative to [1] can be observed:
> >
> > - **LMC barrier approaches zero only at *very high widths*, even for non-Transformer architectures**
> >
> >     If we carefully analyze Figure 4 of [2], we notice that the LMC barrier approaches zero only at rather high [width multipliers](https://github.com/samuela/git-re-basin/blob/main/src/resnet20.py#L79), and importantly, **not for any arbitrary width**. Thus, for typically sized residual or convolutional neural networks on CIFAR10, the LMC barrier in loss is **not zero at all**, and the corresponding barrier when measured in accuracy is even more palpable.
> >
> > - **Transformers possess a more non-convex landscape**
> >
> >     Most importantly, both **[1, 2] only investigate fusion and/or LMC properties on non-transformer architectures**.
> >
> >     The **non-convexity** of the landscape for ViT has been shown [7] to be **much higher** than that of residual networks, which consequently brings about higher LMC barriers. This can also be seen due to the fact that Transformers contain components that further proliferate the number of symmetries, such as within- and across-head permutations as well as the translation invariance due to softmax, — all of which serves to interfere with the linear interpolation of parameters.
> >
> >     Thus, the barriers in [1,2] of non-transformer architecture and those of transformers here are **not** comparable, and **any such comparisons would amount to not accounting for the nature of the problem** being addressed here.

---

> > > ### Author Response · Authors · 2023-11-17
> > > **Rebuttal by Authors 3/3**
> > >
> > > ### Intuitions for soft-alignment
> > >
> > > Soft-alignment inherently interpolates between the case of hard-assignments (low entropy) and completely uniform assignments (high entropy), while minimizing alongside the transportation objective.
> > >
> > > In short, our intuition, which has matured over the course of this work, is that the architectural complications in Transformers aggravate the problem of layerwise neuron alignment, where the soft-alignment essentially allows a **margin of uncertainty in the alignment**.
> > >
> > > In more detail, Transformers are a complicated architecture with multiple information streams (e.g. residual connections, multi-head attentions, and especially the expansion in MLP dimension — more on this below) that were absent in the models fused so far in the literature. All these components, together with the often large size, add **freedom to the pathways and to the internal representation of information inside the architecture**. Hence, one intuition is that these architectural complications clash with the simplistic assumption that representations are encoded exactly in the same layerwise manner across the parent models (i.e., layerwise representation matching), implying that a perfect hard alignment may **not** exist.
> > >
> > > Another aspect that we have observed about the hard-alignment is that it suffers from the particular MLP configuration present in each encoder. In contrast to most of the usual MLP configurations, where the dimensions change smoothly (consistently increasing or decreasing), in Transformers the data is fed into an accordion-like MLP with abrupt projections of the data to and from a much larger space (e.g. of dimensions $d_{in} = 384$, $d_{hidden} = 1536$, $d_{out} = 384$). These projections to a much higher-dimensional space **unconstrain the data representations in that layer, encouraging diversity and even distributed representations** (and thus making it difficult for hard-alignment) between the two models. Here, soft-alignment effectively allows a **margin of uncertainty** in the alignment, which is arguably better than forcing a hard (and possibly wrong) alignment.
> > >
> > > ---
> > >
> > > *We hope that together with the improvements in the manuscript, these further explanations and answers will adequately address your concerns and that — as you have mentioned — you will be able to fully appreciate our work. If however, you might have need of any further clarifications, we remain at your disposal.*
> > >
> > > ---
> > >
> > > ### References
> > >
> > > - [1] Sidak Pal Singh and Martin Jaggi. Model fusion via optimal transport. Advances in Neural Information Processing Systems, 33:22045–22055, 2020
> > > - [2] Samuel K Ainsworth, Jonathan Hayase, and Siddhartha Srinivasa. Git re-basin: Merging models modulo permutation symmetries. arXiv preprint arXiv:2209.04836, 2022
> > > - [3] Jonathan Frankle, Gintare Karolina Dziugaite, Daniel Roy, and Michael Carbin. Linear mode connectivity and the lottery ticket hypothesis. In ICML, volume 119, pp. 3259–3269. PMLR, 2020.
> > > - [4] Mitchell Wortsman, Gabriel Ilharco, Samir Ya Gadre, Rebecca Roelofs, Raphael Gontijo-Lopes, Ari S Morcos, Hongseok Namkoong, Ali Farhadi, Yair Carmon, Simon Kornblith, et al. Model soups: averaging weights of multiple fine-tuned models improves accuracy without increasing inference time. In International Conference on Machine Learning, pp. 23965–23998. PMLR, 2022.
> > > - [5] Frankle et al., Linear Mode Connectivity and the Lottery Ticket Hypothesis, ICML 2020.
> > > - [6] Nagarajan, V. and Kolter, J. Z. Uniform convergence may be unable to explain generalization in deep learning. NeurIPS, 2019.
> > > - [7] [Park & Kim, 2022]: How Do Vision Transformers Work?, ICLR, 2022

---

> > > > ### Author Response · Authors · 2023-11-21
> > > >
> > > > We didn’t mean to bother you, as we understand that you have limited time and many other important occupations. All we want to ask through this comment is that you have a quick, even cursory, glance over our responses, and see if you would still seek additional clarifications.
> > > >
> > > > We thank you again for your constructive feedback and comments, and incorporating those has significantly strengthened our paper, inviting a renewed evaluation.

---

### Official Review · Reviewer_KSPk · 2023-11-01

**Soundness:** 3 good
**Presentation:** 3 good
**Contribution:** 3 good
**Rating:** 6
**Confidence:** 3

**Summary:**

This paper introduces a systematic fusion technique for transformer-based networks by leveraging Optimal Transport to align architectural components. It offers a flexible approach applicable to various architectures, including key Transformer components. Heterogeneous fusion enables efficient compression, with superior performance compared to vanilla fusion and individual parent models, as demonstrated in image classification (Vision Transformer) and natural language tasks (BERT). Our analysis underscores the significance of soft alignment in the context of Transformers, highlighting the potential for combining multiple Transformers to enhance their capabilities in the emerging field of model fusion and recombination.

**Strengths:**

1. The authors examined various strategies (weight vs activation, hard vs soft etc) for applying optimal transport (OT) methods
2. The authors conducted experiments employing both Vision Transformer (ViT) and BERT architectures across multiple datasets.
3. The OT method demonstrates particular efficacy in one-shot scenarios.
4. OT methods exhibit versatility, as they can be effectively applied to models of varying widths, presenting a viable alternative to distillation.

**Weaknesses:**

1. The OT method yields comparatively lower performance when contrasted with ensemble methods.
2. The suitability of the OT method for achieving solid results on larger datasets, such as ImageNet-1K, in one-shot scenarios remains uncertain.

**Questions:**

Please refer to the weakness section.

---

> ### Author Response · Authors · 2023-11-17
> **Rebuttal by Authors 1/2**
>
> *Thanks for your valuable review. We are extremely glad to hear that you have appreciated our systematic investigation of the various fusion strategies. In what follows ahead, we would like to take the opportunity to address your primary concerns.*
>
> ---
>
> ### Accuracy of the fused model
>
> > The OT method yields comparatively lower performance when contrasted with ensemble methods.
>
> Allow us to detail the inherent factors at play when fusing the parameters of the given two neural networks:
>
> - **Non-convexity of the neural network loss-landscape**
>
>   It is widely accepted that neural networks result in a highly non-convex loss landscape and optimization dynamics. As a result, by definition of non-convexity, it is rather unlikely that the loss $\mathcal{L}$ at, say,  the midpoint between two networks (with parameters $\theta_1$ and $\theta_2$) is less than the average of the losses of the two networks. Mathematically,
>
> 	$$ \mathcal{L}\left(\frac{\theta_1 + \theta_2}{2}\right) \nless \frac{1}{2} \mathcal{L}(\theta_1) + \frac{1}{2} \mathcal{L}(\theta_2)\. $$
>
> - **Empirical success of Model Fusion and conjectured Linear Mode Connectivity modulo Permutations**
>
> 	Notwithstanding this inherent non-convexity, [3, 4] showed that, in practice, if the network parameters are aligned prior to fusion, like by accounting for their permutation symmetries, the loss landscape (near the basin of solutions) becomes much less non-convex. The works of [2] and [1] have provided additional evidence that when the network widths are large, the two networks (or modes) can more or less be linearly connected modulo permutation symmetries (RHS - LHS of the above equation is small).
>
>
> - **Recovering the exact permutation is NP-hard**
>
> 	Most algorithmic approaches, either of [3] or [1] are greedy layerwise algorithms. This is because recovering the exact permutation symmetry that links the given networks becomes intractable as the search space grows in proportion to $\mathcal{O}({(m\!)}^L)$, where $m$ denotes the layer width and $L$ is the network depth. Also, see Lemma 1 in [1]. As a consequence, we see the inherent difficulty of the problem that we are tackling here.
>
> - **The added complexity of Transformers**
>
> 	To the best of our knowledge, prior works have only studied fusion and linear mode connectivity mostly in the case of MLPs, CNNs, and ResNets, and **not for Transformers**. On the other hand, Transformers seem to further complicate the above issues in practically finding a good alignment, for instance, due to additional components such as the multi-head self-attention (resulting in the need for intra-head as well as inter-head alignment), LayerNorms, and, more often than not, larger depths. Also, other kinds of symmetries get introduced, such as scale invariance, due to the presence of softmax.
>
> - **Ensembling is implicitly using more capacity**
>
> 	So far we only have outlined the difficulty of the fusion process. However, the other obvious fact is that ensembling uses $K \times$ more capacity than fusion, where $K$ is the number of networks that are ensembled. Hence, the ensembling performance can be regarded, in some sense, as the unachievable upper bound on the performance of the singular network produced through fusion.
>
>
> **In this light, arguably, it is rather noteworthy that like prior works:**
>   1. We can still get highly non-trivial performance gains over vanilla fusion. In the case of Transformers, we believe this is also where soft alignment benefits since it helps relax the constraints from hard alignment (permutation matrices) to softer alignments (with transportation maps).
>   2. Moreover, after fine-tuning, even this **gap relative to ensembling considerably diminishes**. Besides, this fine-tuning procedure is usually extremely fast — requiring as few as **0.4%** of the original training duration —, and since fine-tuning has to be done just once even this minor cost gets amortized over inference (while ensembling must pay $O(n)$ (where $n$ is the number of parent models) more at every inference call)

---

> > ### Author Response · Authors · 2023-11-17
> > **Rebuttal by Authors 2/2**
> >
> > ### One-shot fusion
> >
> > > The suitability of the OT method for achieving solid results on larger datasets, such as ImageNet-1K, in one-shot scenarios remains uncertain.
> >
> > Our focus for the experiments on ImageNet-1k was on the finetuning performance, carrying our intuition from smaller datasets about the short finetuning phase capable of outperforming both parent models. This intuition indeed proved to be correct (Tab. 3). We remark that our OT method always delivers far better one-shot performance than VF, and even here for ImageNet-1K, but we chose to concentrate on finetuning performance:
> >
> > 1. The one-shot performance on ImageNet-1k has not been tuned as we simply took the hyperparameters from smaller dataset settings, where ablations were computationally feasible (Tab. 1 and Fig. 5).
> > 2. Besides, for ImageNet-1K, it can be seen in Ainsworth et al. (2022) that there is a larger loss barrier (due to the inherently more non-convex nature of the problem), which makes fine-tuning — a procedure which will anyway be carried out in practical scenarios — necessary.
> >
> > Importantly, with finetuning, as extensively discussed (Tab. 2, 3, 4), our method delivers compelling results, with the fused model consistently outperforming VF and even both parent models in both same-size and heterogeneous fusion.
> >
> > ---
> >
> > *We hope that our detailed response above helps resolve your concerns. In case you might have further questions or comments, we remain at your disposal.*
> >
> > ---
> >
> > ### References
> > - [1] Samuel K. Ainsworth, Jonathan Hayase, and Siddhartha Srinivasa. Git Re-Basin: Merging Models modulo Permutation Symmetries. *International Conference on Learning Representations*, 2022.
> > - [2] Rahim Entezari, Hanie Sedghi, Olga Saukh, and Behnam Neyshabur. The Role of Permutation Invariance in Linear Mode Connectivity of Neural Networks. *International Conference on Learning Representations*, 2022.
> > - [3] Sidak Pal Singh and Martin Jaggi. Model fusion via optimal transport. *Advances in Neural Information Processing Systems*, 33:22045–22055, 2020.
> > - [4] Hongyi Wang, Mikhail Yurochkin, Yuekai Sun, Dimitris Papailiopoulos, and Yasaman Khazaeni. Federated Learning with Matched Averaging. *International Conference on Learning Representations*, 2020.

---

> > > ### Author Response · Authors · 2023-11-21
> > >
> > > We didn’t mean to bother you, as we understand that you have limited time and many other important occupations. All we want to ask through this comment is that you have a quick, even cursory, glance over our responses, and see if you would still seek additional clarifications.
> > >
> > > We thank you again for your constructive feedback and comments, and incorporating those has significantly strengthened our paper, inviting a renewed evaluation.

---

> > > > ### Comment · Reviewer_KSPk · 2023-11-22
> > > >
> > > > Thank you for your reply. I will maintain the existing score

---

### Official Review · Reviewer_5GYY · 2023-11-01

**Soundness:** 2 fair
**Presentation:** 2 fair
**Contribution:** 3 good
**Rating:** 6
**Confidence:** 4

**Summary:**

This paper proposed a systematic approach for fusing two or more pretrained transformers by studying the flow of transportation maps in each specific component of Transformer. The authors empirically showed that when working with Transformers, hard alignment underperforms soft alignment in one-shot fusion, which is in contrast to the cases of fully connected and convolutional neural networks. Finally, they showcased the efficiency of the proposal in fusing and finetuning ViT and BERT.

**Strengths:**

- This paper is well-structured.
- To the best of my knowledge, this is the first work that aims to fuse transformer architectures by aligning their weights.
- The proposed method is successfully backed by theoretical results.

**Weaknesses:**

- The methodology part is not well-written and lacks some details.

**Questions:**

- Section 2: The model fusion literature has some papers that are slightly off: Tatro et al., 2020; Juneja et al., 2022; Kandpal et al., 2023.
- Eq. 2: What is $f$? What is its output?
- Section 4.2.1: How to calculate weighted matrix?
- The authors should remind the formulation for Attention operation in either Section 3 or Section 4.2.2.
- Section 4.2.2:
  - Where do the authors remove the constraints in Section 4.2.2?
  - It is unclear how to calculate $T_Q$ and $T_K$. Did the authors check the assumption $T_Q = T_K$ in the experiments?
  - What are $W_i^Q, W_i^K$, and $W_i^V$? Does $i$ indicate the head index?
  - Additional visualizations may help to demonstrate the method here.
- Section 4.2.3: What is this sentence for? “For the concatenation, we notice that the class token is only a small fraction of the full sequence, in other words, for the integrity of the sequence, it is far more important to propagate the TM of the patch embeddings than the one for the class token.” In addition, the class token is more important because it gathers the information from the patch.


**Minors**:
- Eq. 3 should be moved up a paragraph.

---

> ### Author Response · Authors · 2023-11-17
> **Rebuttal by Authors 1/1**
>
> *We sincerely appreciate your careful reading of our paper and giving constructive feedback. We are pleased to hear that you find our paper well structured and that you have appreciated the novelty of our work. We have seriously considered your notes and we recognize that some crucial aspects could have lacked clarity.*
>
> *Owing to your insightful feedback we have implemented various modifications in the methodology section (Sec. 4) of the manuscript that, in our opinion, have improved the overall comprehensibility of our work.*
>
> ---
>
> ### Related Work
>
> > “[…] some papers that are slightly off: Tatro et al., 2020; Juneja et al., 2022; Kandpal et al., 2023.”
>
> We have fixed these.
>
> ---
>
> ### Residuals (Sec. 4.2.1)
>
> > “Eq. 2: What is $f$? What is its output?”
>
> $\mathbf{f}_{residual}$ is a vector and stands for the activations coming from the residual branch.
>
> $\mathbf{f}_{current}$ is also a vector and stands for the activations coming from the current layer l. We added a sentence for clarification in the paper and adjusted Eq. 2 to make it more clear.
>
> > “How to calculate weighted matrix?”
>
> The calculation for the weighted matrix is very similar to the weighted scalar. The only difference is that, for weighted matrix, we compute a weighting factor for every incoming residual strand, instead of one common value. We added a sentence for clarification in the paper.
>
> ---
>
> ### Multi-Head Attention (Sec. 4.2.2)
>
> > “[...] remind the formulation for Attention operation [...]”
>
> Done
>
> > “Where do the authors remove the constraints in Section 4.2.2?”
>
> We assume you mean the “equal transportation map” constraint here. Section 4.2.2 serves to explain how we handle the transportation map flow through the self-attention block. At the beginning of the section, we describe our analytical insight on why one should only propagate $\mathbf{T_V}$. Eq. 4 shows that if we use $\mathbf{T_Q} = \mathbf{T_K}$, these permutation matrices cancel. However, this equation is not valid anymore in the case of soft-alignment because we no longer have permutation matrices, so the $\mathbf{T_Q} = \mathbf{T_K}$ constraint is no longer needed. We added a sentence for clarification in the paper.
>
> > “It is unclear how to calculate $\mathbf{T_K}$ and $\mathbf{T_Q}$”
>
> If we do not assume the constraint on the outgoing transportation maps $\mathbf{T_K} = \mathbf{T_Q}$ one can directly apply OTFusion to $\mathbf{W_K}$ and $\mathbf{W_Q}$ (as for any fully-connected layer). OTFusion yields $\mathbf{T_K}$ and $\mathbf{T_Q}$ as the outgoing transportation map respectively.
>
> In the case of $\mathbf{T_K} = \mathbf{T_Q}$, we compute one common ground metric for both $\mathbf{W^K}$ and $\mathbf{W^Q}$. In the case of activation based alignment the ground metric is computed from their combined activations. For weight based alignment, the ground metric is computed from the concatenation of $\mathbf{W^Q}$ and $\mathbf{W^K}$. From this common ground metric we then compute a common transportation map that we use to align both $\mathbf{W^K}$ and $\mathbf{W^Q}$.
>
> > “What are $\mathbf{W^K_i}$, $\mathbf{W^Q_i}$, $\mathbf{W^V_i}$, and ? Does i indicate the head index?”
>
> Yes, $i$ indicates the head index. $\mathbf{W^K_i}$, $\mathbf{W^Q_i}$, and $\mathbf{W^V_i}$ are the corresponding weight matrices. We added a clarification in the manuscript, and together with the inserted self-attention operation formulation, it should be clear now.
>
> ---
>
> ### Embeddings (Sec. 4.2.3)
>
> > “What is this sentence for? “For the concatenation, we notice that the class token [...]” In addition, the class token is more important because it gathers the information from the patch.”
>
> We assume that at the very beginning of the architecture, namely at the embedding stage of the transformer, each patch and token is of similar importance. The intuition for this assumption is that we concatenate only one class token to tens of image patches, and we can therefore assume that it is more important to propagate the transportation map of the image patches instead of the class token.
>
> Note however that we make this assumption **exclusively for the embeddings**, i.e. **not throughout all encoder blocks** where, as you say, the information will indeed be eventually distilled into the class token.
>
> ---
>
> *We hope that in light of the modifications of the manuscript and the above clarifications, we have adequately addressed your questions and concerns. In case you might have further doubts or comments, we remain at your disposal.*

---

> > ### Author Response · Authors · 2023-11-21
> >
> > We didn’t mean to bother you, as we understand that you have limited time and many other important occupations. All we want to ask through this comment is that you have a quick, even cursory, glance over our responses, and see if you would still seek additional clarifications.
> >
> > We thank you again for your constructive feedback and comments, and incorporating those has significantly strengthened our paper, inviting a renewed evaluation.

---

> > > ### Comment · Reviewer_5GYY · 2023-11-22
> > > **Respone to authors' rebuttal**
> > >
> > > I thank the authors for patiently reading and responding to all my questions. I found that my concern was adequately addressed.
> > >
> > > I would like to maintain my score and incline to an acceptance.

---

### Official Review · Reviewer_GM8g · 2023-11-01

**Soundness:** 3 good
**Presentation:** 3 good
**Contribution:** 3 good
**Rating:** 6
**Confidence:** 4

**Summary:**

This paper proposes a systematic approach for fusing two or more transformer-based networks exploiting Optimal Transport technique. The proposed method can generalize to arbitrary architectures for CNNs and Transformers. Extensive experiments involving the fusion and finetuning of Vision Transformers (ViTs) across multiple datasets demonstrate the effectiveness of the proposed method.

**Strengths:**

- This paper is well written.
- The proposed method shows good generalization across different architectures.
- The proposed method show strong performance for several benchmark.

**Weaknesses:**

- Most experiments are conducted to compare with Vanilla Fusion. More comparisons with state-of-the-art methods should be included.
- Most experiments are conducted on CIFAR dataset which is relatively small.

**Questions:**

See the weakness part.

---

> ### Author Response · Authors · 2023-11-17
> **Rebuttal by Authors 1/2**
>
> *Thank you for your useful feedback. We are pleased to hear that you have found our paper well written, and have appreciated the generalization capabilities of our method across different architectures.*
>
> ---
> > "Most experiments are conducted to compare with Vanilla Fusion. More comparisons with state-of-the-art methods should be included."
>
> In general, we share a similar sentiment as you have expressed. We would have liked to add more comparisons, however, we were forced to compare primarily with Vanilla Fusion (with and without fine-tuning; as well as parent models and ensemble), as **we are not aware — to the best of our knowledge — of any other method that can handle the problem of aggregating the parameters of multiple transformer models, in particular, when they are distant in their parameter space**.
>
>
> More specifically, to better contextualize the state-of-the-art (SotA) in this area, we would like to reiterate here some fundamental differences and limitations of notable related work:
>
> - **OTFusion [1]**
>   - [1] first introduces the concept of Optimal Transport based alignment and fusion. However, its fully layerwise interpretation **lacks generalization capabilities**, and as such it is only applicable to simple architectures such as multi-layer perceptrons, CNNs, and instances of ResNet.
>   - It is **not equipped in any way to align and fuse models with complex information streams** and to fuse transformer-specific components such as multi-head attention layers, layer-normalization, embeddings, or the sequential nature of the data.
> - **Successors of OTFusion**
>   - A plethora of other methods emerged from the ideas and methodology of [1] and explored various adaptations and different applications.
>   -  In particular,  [2] focused on the linear mode connectivity perspective for MLPs, CNNs, and ResNets, while [3] focused on RNNs and LSTM architectures. These methods, too, are **not applicable to transformers** and a direct comparison is consequently out of reach.
> - **Model Soups [4] for transformers**
>   - Recently, [4] focused on weight averaging specifically for transformers, reaching **SotA performance**.
>   - The inherent methodology of [4] actually **relies on VF itself, namely one-to-one averaging** of the parameters. However, there is a subtle but essential difference compared to our application scenario: their parent models originate from the same pre-trained model and therefore the parent models remain **sufficiently close** in the parameter space. This precludes the need to align them and lets them employ a simple averaging strategy while still obtaining a gain in performance.
>   - For this reason, **this method does not apply to our problem where we fuse transformer networks that are potentially much more distant in their parameter spaces** and are diverse in nature.
>
> Furthermore, we would like to stress that the larger point of our paper is to present the **first successful model fusion technique for transformers that aggregates their parameters**, with a focus on modularity, generalization capabilities, and systematic investigation of fusion of the transformers' peculiar components and characteristics (such as multi-head self-attention, layer normalization, sequentiality of the data, to name a few).
>
> ---
>
> > "Most experiments are conducted on CIFAR dataset which is relatively small."
>
> We agree that CIFAR10 is a relatively small dataset, and we opted for this dataset for the experiments and ablations (Tab. 1, Fig. 4 and 5) that required multiple evaluations with different hyperparameters and fusion strategies, and that were prohibitive with large datasets. However, sharing your concern, and aiming to demonstrate the wide applicability and generalization capabilities of our method, we also have **experiments on other larger datasets**.
>
> ### ViT
>
> In particular, we have presented the results of ViT fusion also on:
>
> - **CIFAR100** (Tab. 2 for models of the same size; Tab. 4 for heterogeneous fusion)
> - **TinyImageNet** (Tab. 15 in the Appendix),
> - **ImageNet-1k** (Tab. 3)
>
> where we see a consistent gain across all settings.
>
> ### BERT
>
> Furthermore, we would like to highlight that not only have we applied our method to the ViT, but **also to BERT for NLP tasks, evaluated on the widely adopted GLUE benchmark** (Tab. 17, in the Appendix), and the results were similar to those for the vision tasks.
>
> ---
>
> *All in all, we hope that in light of the above clarifications, we have given a proper answer to your concerns. Should you have any further questions, or comments, we will be more than happy to answer them. Thanks for taking the time.*

---

> > ### Author Response · Authors · 2023-11-17
> > **Rebuttal by Authors 2/2**
> >
> > ### References:
> > - [1] Sidak Pal Singh and Martin Jaggi. Model fusion via optimal transport. Advances in Neural Information Processing Systems, 33:22045–22055, 2020
> > - [2] Samuel K Ainsworth, Jonathan Hayase, and Siddhartha Srinivasa. Git re-basin: Merging models modulo permutation symmetries. arXiv preprint arXiv:2209.04836, 2022
> > - [3] Aditya Kumar Akash, Sixu Li, and Nicolás García Trillos. Wasserstein barycenter-based model fusion and linear mode connectivity of neural networks. arXiv preprint arXiv:2210.06671, 2022.
> > - [4] Mitchell Wortsman, Gabriel Ilharco, Samir Ya Gadre, Rebecca Roelofs, Raphael Gontijo-Lopes, Ari S Morcos, Hongseok Namkoong, Ali Farhadi, Yair Carmon, Simon Kornblith, et al. Model soups: averaging weights of multiple fine-tuned models improves accuracy without increasing inference time. In International Conference on Machine Learning, pp. 23965–23998. PMLR, 2022.

---

> > > ### Author Response · Authors · 2023-11-21
> > >
> > > We didn’t mean to bother you, as we understand that you have limited time and many other important occupations. All we want to ask through this comment is that you have a quick, even cursory, glance over our responses, and see if you would still seek additional clarifications.
> > >
> > > We thank you again for your constructive feedback and comments, and incorporating those has significantly strengthened our paper, inviting a renewed evaluation.

---

> > > > ### Comment · Reviewer_GM8g · 2023-11-22
> > > > **Thanks for the response**
> > > >
> > > > Thanks for the detailed response from the authors. Overall I am still not quite sure about the claims that no prior methods can be compared. More experiments on ImageNet are highly encouraged. Therefore I tend to keep my original scores.

---

> > > > > ### Author Response · Authors · 2023-11-23
> > > > >
> > > > > *Thanks for answering and for the further feedback.*
> > > > >
> > > > > ---
> > > > >
> > > > > > "Overall I am still not quite sure about the claims that no prior methods can be compared."
> > > > >
> > > > > We are **not** aware of any other method or approach to fuse transformer models diverse in their parameter space, other than simple one-to-one vanilla averaging (Model Soups [3]).
> > > > >
> > > > > Nevertheless, and to not fall short of your expectations, we present comparisons of our work with some baselines that are based on relevant prior work that **did not aim at fusing transformers**. Therefore, these works do not offer any insight into transformer-specific components (multi-head attention, layer normalization, etc.) and how to fuse them; we therefore handle these components with the only method we are aware of, i.e., Model Soups (VF). The results highlight the breakdown of previous work and thus the importance of our contribution to transformer fusion at large.
> > > > >
> > > > > The following results are evaluated on the same models as in Tab. 1 of our manuscript.
> > > > >
> > > > > | Parent Models| VF (Model Soups [3]) | OT acts. [1]* | OT wts. [1]* | OT + [2]'s resid. ($\mathbf T_{out} = \mathbf T_{resid}$)* | OT + [2]'s resid. ($\mathbf T_{out} = \mathbf I$)* | OT-acts (ours) | OT-wts (ours) |
> > > > > |--- |------------------|--------------------------|----------------------|---------------------------------------------|----------------------------------------|----------------|--------------------|
> > > > > | [92.3%, 92.3%] | 7.6%            |                   10.9% |               12.9% |                                      11.4% |                                 10.5% |      **60.2%** | **57.8%**             |
> > > > >
> > > > > *: Baselines without our contributions, i.e. hard-alignment fusion, and VF of transformer-specific components (not handled by those papers).
> > > > >
> > > > > ---
> > > > >
> > > > > > "More experiments on ImageNet are highly encouraged."
> > > > >
> > > > > Furthermore, as a last-minute addition, we present a **further result on ImageNet-1k**. Since training from scratch two further ViT on ImageNet-1k is computationally expensive (even more within the rebuttal period), we have instead used off-the-shelf models. In particular, we have used the "vit-base-patch16-224-in21k” from Google (size: [768, 3072, 12]), and we have subsequently finetuned this pre-trained network on ImageNet-1k with two different seeds to get two parents for our fusion method.
> > > > >
> > > > > With two parent models of accuracies 79.9% and 79.8%, the model fused with our method achieves without finetuning a performance of 81.3%. However, we note that given the parent networks start off from the same base network, they remain close in their parameter space, and it is therefore a less challenging task than the experiments presented in Tab. 1, 2, 3, and 4 of our paper. Also, for the same reason, we expect an approach like Model Soups to be not too different.
> > > > >
> > > > > ---
> > > > >
> > > > > *We thank you for engaging in the discussion, and we hope that these further results helped address your remaining doubts.*
> > > > >
> > > > > ---
> > > > >
> > > > > ### References
> > > > >
> > > > > - [1] Sidak Pal Singh and Martin Jaggi. Model fusion via optimal transport. Advances in Neural Information Processing Systems, 33:22045–22055, 2020
> > > > > - [2] Samuel K Ainsworth, Jonathan Hayase, and Siddhartha Srinivasa. Git re-basin: Merging models modulo permutation symmetries. arXiv preprint arXiv:2209.04836, 2022
> > > > > - [3] Mitchell Wortsman, Gabriel Ilharco, Samir Ya Gadre, Rebecca Roelofs, Raphael Gontijo-Lopes, Ari S Morcos, Hongseok Namkoong, Ali Farhadi, Yair Carmon, Simon Kornblith, et al. Model soups: averaging weights of multiple fine-tuned models improves accuracy without increasing inference time. In International Conference on Machine Learning, pp. 23965–23998. PMLR, 2022.

---

### Author Response · Authors · 2023-11-17
**Author Rebuttal**

*We would like to thank all reviewers for taking the time to read our paper and giving insightful feedback. We truly appreciate the unanimous consensus towards recommending the acceptance of our paper.*

*We are excited to hear that reviewers GM8G, 5GYY and KSPk have appreciated the contribution of our work. We are also glad to hear that hByk has appreciated the clarity of our paper, and found it very well-written.*

---

To address the questions and doubts raised by the reviewers, we have modified our manuscript as follows and noted all changes in the PDF in pink color.

* Following feedback from reviewer 5GYY we have modified the methodology section (Sec. 4) to **improve the clarity of our method**.

  In particular, we have expanded the description of multi-head attention fusion by adding additional formulas and an explicative visualization (Fig. 6 in Appendix B).

  Also, we have improved the notation of Eq. 2.

* As suggested by hByk, we have further **highlighted our contribution**, focusing on distinguishing between [1] and our work.

  To this end, we have made minor adjustments throughout the entire work. Specifically, in Sec. 1 and Sec. 2 we have explicitly reminded OTFusion limitations and the impossibility to fuse Transformers, while in Sec. 4.3 we have better conveyed the need for soft-alignment for Transformers compared to simpler architectures.

---
*We thank all the reviewers for their useful feedback, as a result of which we have been able to further strengthen our paper.*

---

### Meta-Review · Area_Chair_kymv · 2023-12-06

**Metareview:**

This paper proposes a method to fuse transformer models using an Optimal Transport technique to align their weights. The key idea, proposed in (Singh et al. 2020), is to align the rows/columns of weight matrices using a soft-matching obtained by means of solving a (Kantorovich) Optimal Transport problem. This idea is extended in this paper to align the building blocks of Transformers such as multi-head self-attention, layer-normalization, and residual connections, and can be used to fuse models of different sizes. Experimental results on Vision and Language Transformers show that this approach outperforms vanilla fusion in most cases.

The main concerns raised by reviewers related to the contribution of the paper (especially with respect to Singh et al. 2020), the weakness of the baseline (Vanilla fusion) and the limited scope of the experiments. Regarding novelty, it should be noted that this is neither the first method that fuses transformer architectures through weight averaging (see eg Dansereu et al. 2023), neither the first method to use OT for weight alignment (e.g., Singh et al. 2020 and many others after that). It does seem to be the first paper that combines these two ideas. The other two points seem to have been mostly addressed in the rebuttal phase, which at least one reviewer increasing the score.

Overall, it seems that this paper, despite its limited novelty, contains insights and results that might be interesting enough for the community as to warrant acceptance to the conference.

**Justification For Why Not Higher Score:**

The marginal contribution of this paper with respect to the prior work upon which it builds (Sing et al. 2020) is limited.

**Justification For Why Not Lower Score:**

The paper is sound, well-written, and contains some interesting insights that might be valuable to the community.

---

### Decision · Program_Chairs · 2024-01-16

Accept (poster)